Journal of Data-centric Machine Learning Research (2026)        Submitted 8/25; Revised 10/25; Published 2/26

# Automated Data Preparation for Machine Learning: A Survey

**Sasa Mladenovic**                                                    SASA.MLADENOVIC@LIP6.FR
*Sorbonne Université, CNRS, LIP6*
*Paris, France*

**Marius Lindauer**                                          M.LINDAUER@AI.UNI-HANNOVER.DE
*Leibniz University Hannover, L3S Research Center*
*Hannover, Germany*

**Carola Doerr**                                                       CAROLA.DOERR@LIP6.FR
*Sorbonne Université, CNRS, LIP6*
*Paris, France*

**Reviewed on OpenReview:** *https: // openreview. net/ forum? id= Euti6LHIOs*

**Editor:** Peter Mattson

## Abstract

Data preparation is essential for effective machine learning (ML), yet typically remains a manual, time-consuming process. While automated machine learning (AutoML) has successfully addressed modeling aspects of the ML workflow, data preparation has largely been overlooked, leading to challenges with real-world, imperfect data. Conversely, a rising paradigm in the world of artificial intelligence (AI) and ML is that of data-centric AI, shifting focus from just refining models, to enhancing data in order to advance performance boundaries. This survey motivates the need for automated solutions regarding data preparation, offering a fundamental understanding of the benefits of data transformations and establishing the complexity of data pipeline optimization, while highlighting the importance of data quality. We provide a comprehensive overview and categorization of existing automation approaches, both in AutoML and as standalone fully or semi-automated systems. We discuss underlying methodologies, their advantages, and limitations. Our work explores the prospects of expanding automation to cover a broader data preparation process, aiming to bridge the gap between data-centric AI and AutoML. It paves the way to a wholly automated pipeline from raw real-world data to quality model predictions, and outlines future research directions towards that goal.

**Keywords:** Automated data preparation, data preprocessing, data pipeline optimization, machine learning, data quality, AutoML.

## 1 Introduction

*Motivation.* The field of artificial intelligence (AI), more specifically machine learning (ML), has undergone a period of explosive growth in recent years. This phenomenon has been fueled by multiple factors, including increasing availability of large amounts of data, advancements in computational power driven by hardware improvements and cloud computing, as well as significant algorithmic innovations such as novel neural network architectures, improved training strategies, and more efficient optimization techniques. One of the challenges that has been raised among these innovations is that of *automating data science* (Bouneffouf et al., 2020; De Bie et al., 2022).

Data science (DS) allows us to extract insights about the real world trough data-driven approaches, which often draw upon ML techniques. Machine learning heavily relies on human expertise, namely knowledge of the domain its data stems from, as well as technical skills in mathematics, programming, and ML itself. Nonetheless, a significant part of its workflow can also be delegated to automation. This has been the focus of the domain of automated machine learning (AutoML) (Hutter et al., 2019), which leverages automation techniques with the objectives of widening the machine learning audience by making ML more accessible to non-experts, improving the efficiency of the ML process via optimized design choices, and accelerating research in the field through faster experimentation.

The machine learning pipeline can be subdivided into two overarching components: (1) data preparation and (2) modeling. *Data preparation* designates the application of a sequence of transformations to raw data. Its goal is to ensure usability of the data by resolving any incompatibilities with a given ML model, and optimize result quality once it is input into that model. *Modeling*, which represents the core of the ML process, encompasses selecting the most adequate machine learning algorithm to apply to the data, and tuning its hyperparameters so as to obtain the best possible outcomes. These two components are closely intertwined: data preparation can be tailored to a chosen model, while at the same time the state of the data influences the best choice of model (Oala et al., 2024).

The field of data science incorporates a deeper data preparation segment that reaches beyond typical data treatment in ML. While the latter is largely focused on model compatibility and performance, broader data preparation, as seen by the data management (DM) field, extends to processes aimed at organizing data and raising its quality (Chai et al., 2023b). These aspects serve to improve the efficiency, effectiveness, and reliability of any downstream analytics and systems, including machine learning pipelines.

In this work, we consider a holistic data preparation definition integrating the understanding of the term by both the data management and machine learning domains, as both eventually have a significant impact on ML outcomes. Figure 1 illustrates the structure and relationships between the different entities and processes composing a DS workflow that enfolds an ML pipeline. The figure highlights the data preparation process, and outlines its interaction mechanisms within the larger ML and DS workflows.

Both the data preparation and modeling steps of the machine learning pipeline are essential for good results (Lawrence, 2017; Aroyo et al., 2022; Oala et al., 2024), in terms of model

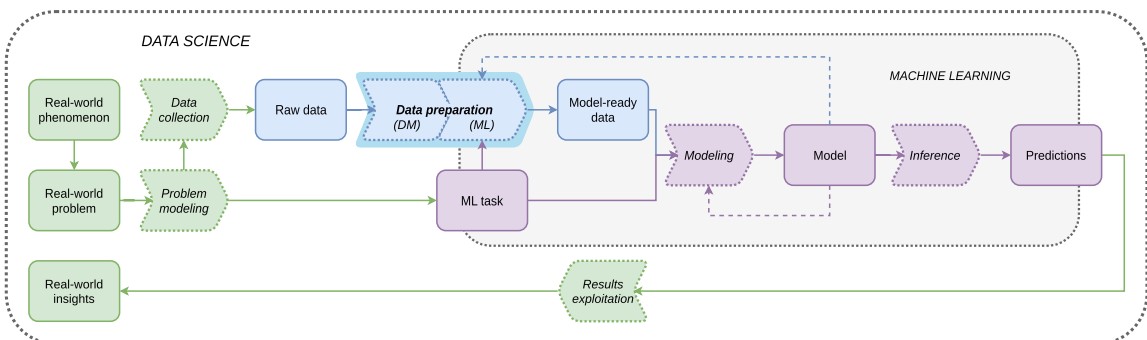

Figure 1: Data preparation within the ML and DS workflows

quality as well as time and memory performance. In some conditions, it may be possible to overcome certain types of data imperfections with robust models instead of resorting to data preparation (Picado et al., 2020; Cvetkov-Iliev et al., 2022; Neutatz et al., 2022). However, this is not the general case. In practice, it is common for data preparation to occupy the majority of a data scientist or ML practitioner's time, with literature estimating shares of 50-90%, and multiple claims of around 80% (Munson, 2012; Bilalli et al., 2016; Shah and Kumar, 2019; Whang and Lee, 2020; Kumar et al., 2024). The customary manual treatment of both data and model optimization is in fact quite time-consuming, typically warranting an iterative trial and error approach by an expert in ML or data science—hence the appeal of AutoML, which reduces the human effort and bias involved in these slow and repetitive tasks, while simultaneously optimizing the performance of the ML system.

Historically, the machine learning field has been dominated by model-centric approaches, concentrated on achieving performance by optimizing algorithms, models, and hyperparameters. While the data management field has long focused on improving data systems, relatively little of that work has actually been adopted in ML workflows. Yet, data quality often presents a bottleneck in ML (Singh, 2023; Malerba and Pasquadibisceglie, 2024), as models are ultimately only as effective as the data that shapes them (Jain et al., 2020; Whang et al., 2023; Zha et al., 2023). This is particularly true in real-world ML applications, where data is typically imperfect in many ways (e.g., inconsistent, noisy, incomplete), thus termed *dirty data* (Hernández and Stolfo, 1998; Kim et al., 2003). Recognizing not only this setback, but also the potential performance upside of optimizing data (Bhatt et al., 2024; Malerba and Pasquadibisceglie, 2024), the rising data-centric AI (DCAI) (Jarrahi et al., 2023; Zha et al., 2023; Jakubik et al., 2024; Kumar et al., 2024; Zha et al., 2025) paradigm shifts the focus from refining the model to enhancing the data. By prioritizing data quality, data-centric machine learning research (DMLR) (Oala et al., 2024) aims to bring the added effectiveness, efficiency, and reliability of data-centric approaches to ML processes.

Taking a closer look at AutoML research, we observe strong tendencies favoring the modeling phase of the ML pipeline, while the data preparation aspect is mostly overlooked, essentially following the same trend as traditional machine learning (Oala et al., 2024). This imbalance is witnessed by scarce publications around automating data preparation, in addition to little attention to the topic in AutoML surveys (Karmaker , "Santu"; He et al., 2021; Zöller

and Huber, 2021; Baratchi et al., 2024; Salehin et al., 2024; Shen et al., 2024). In fact, leading openly available AutoML systems often necessitate already preprocessed input data or provide minimal data preparation (Truong et al., 2019; Krauß et al., 2020; Krauß, 2020; Bilal et al., 2022), whilst focusing on modeling, which can significantly limit their usability on real-world data (Renggli et al., 2022).

Moreover, there appears to be a disconnect between claims that AutoML and resulting industry solutions have solved the problem of automating the entirety of the ML workflow, and the reality of remaining data issues and lack of meaningful benchmarks for real applications (Kumar, 2021)—thus raising the question of *automating data preparation*, and prompting inquiry into its feasibility, usefulness, and necessity (Paton, 2019).

*Objective.* In this article, we survey the existing body of knowledge on automating the data preparation process in the context of machine learning, seeking to offer insights regarding scope, relevance, challenges, and methodologies. We lay the groundwork for research on the holistic integration of data preparation into automated machine learning, in an effort to facilitate extending the reach of automation to the complete data science workflow, and bridge the gap between AutoML and data-centric AI.

*Scope.* We delve into the specifics of data preparation, covering the entire data treatment pipeline between raw source data and fully processed data as it is used for ML model input. While many different terms are frequently employed in literature to designate parts of this process—data wrangling, data engineering, data cleaning, data preprocessing, feature engineering, and several others—their meanings often vary or overlap, hence our use of the term *data preparation* to unify them.

In an overview of underlying data transformations, we direct our attention to the different categories of operations and their value to the overall process. We strive to provide a well-rounded and structured approach with representative examples. We further expand upon the challenges in designing and evaluating data pipelines.

As the core of our work, we survey developments around the topic of automating the data preparation process. In particular, we examine the presence and role of data transformations in existing AutoML systems. Additionally, we explore the numerous and varied approaches to automating data preparation. This includes approaches covering the complete process or specific data aspects, operating in a fully automated manner or aided by human input.

ML data is characterized by its modality, i.e., the nature or source of the data (Baltrušaitis et al., 2019; Parcalabescu et al., 2021; Borisov et al., 2024), as well as its format, or structural representation. Data science is largely centered around tabular data, with data points as rows, and features of varying data types as columns. This has greatly influenced the growth of AutoML, where tabular is the most widely supported data modality, but the field has also extended to others, such as time series, text, or images. These trends are reflected in our study, in particular in surveyed systems, and our examples of data transformations have been chosen accordingly. Nonetheless, we include instances of other data modalities among

both, in order to constitute a more complete overview. More importantly, automated data preparation methodology aspects of our work are *a priori* independent of data modality.

*Contributions.* To the best of our knowledge, this survey is the first of its kind to focus on data preparation for machine learning in its entirety, and specifically on the different facets of its automation. Our main contributions lie in establishing a thorough background for the complete data preparation process, proposing a taxonomy of the data transformations within, exploring challenges, and providing a review of the current state of research on automating the process—thus creating a connecting thread from fundamentals of the field, through current automation progress, to prospects for further advancement. To be more precise:

1. We initially present a comprehensive background of data preparation in ML and the relevant context. We highlight the different categories of data transformations and their most common operations, and provide a meaningful structure according to their roles. We assess the challenges in data preparation pipeline design, as well as in their evaluation.

2. To begin investigating automation aspects, after compiling a list of pertinent AutoML systems, we group them them by data pipeline optimization architecture. We deliver an analysis of the automated data preparation elements integrated into each system, and extract common tendencies.

3. We then examine semi-automated data preparation frameworks, taking into account scope, features, and objectives, and categorizing them with regard to their varied approaches. We further derive insights on the feasibility and challenges of automating the whole process.

4. Following that, we provide an overview of fully automated data preparation methodologies, with example systems belonging to each. We review their merits and shortcomings, and present a comparative analysis along relevant criteria for the choice of quality solutions.

5. Lastly, we summarize and discuss our findings, and consider future research directions.

*ML Terminology.* In order for the paper to be self-contained, we provide definitions of ML-related terms relevant to the context. They can be found in Table 5 in Appendix Section A.

## 2 Related Work

Despite the lack of comprehensive studies on holistic automated data preparation for machine learning, multiple recent works provide meaningful views into the topic or certain subareas. These include data science surveys (Bouneffouf et al., 2020; De Bie et al., 2022) dedicating a portion of their attention to data preparation aspects, with consideration for different systems or methods. There are also data preparation (Salhi et al., 2023) or AutoML (He et al., 2021; Zöller and Huber, 2021; S and Battineni, 2023; Salehin et al., 2024) surveys, AutoML benchmarks (Truong et al., 2019; Krauß, 2020; Krauß et al., 2020) and other works (Bilal et al., 2022) which consider data preparation. Finally, two very recent surveys

specifically revolve around automated data preparation (Mumuni and Mumuni, 2025)—but with areas of focus different from ours—and on the usability of automated feature engineering systems (Schäfer et al., 2025).

Table 1 presents an overview of these works, along with a comparison of their contributions in relation to ours. It indicates (1) whether the research is in the form of a survey paper, and (2) whether its primary focus is on data preparation. It then outlines which major facets of automated data preparation the research deals with, by observing (3) analyzed systems and methods, as well as (4) whether the data preparation is purely ML model-focused or also includes considerations for data organization and quality.

| Reference(s) | Is survey paper | Is focused on auto. DP | Includes analysis of: | | | Considers data: | |
|---|---|---|---|---|---|---|---|
| | | | DP in AutoML sys. | Auto. DP sys. | DP opt. methods | Organization & quality | Opt. for ML model |
| (Truong et al., 2019) | - | - | ✓ | - | - | - | ✓ |
| (Krauß et al., 2020; Krauß, 2020) | - | - | ✓ | - | - | - | ✓ |
| (Bilal et al., 2022) | - | ✓ | ✓ | - | - | - | ✓ |
| (Bouneffouf et al., 2020) | ✓ | - | - | - | ✓ | ✓ | ✓ |
| (He et al., 2021) | ✓ | - | - | ✓ | - | ✓ | ✓ |
| (Zöller and Huber, 2021) | ✓ | - | - | ✓ | ✓ | ✓ | ✓ |
| (De Bie et al., 2022) | ✓ | - | - | ✓ | - | ✓ | ✓ |
| (S and Battineni, 2023) | ✓ | - | ✓ | - | - | - | ✓ |
| (Salehin et al., 2024) | ✓ | - | - | ✓ | - | ✓ | ✓ |
| (Salhi et al., 2023) | ✓ | ✓ | ✓ | - | - | ✓ | ✓ |
| (Mumuni and Mumuni, 2025) | ✓ | ✓ | - | ✓ | - | ✓ | ✓ |
| (Schäfer et al., 2025) | ✓ | ✓ | - | ✓ | - | - | ✓ |
| **Our work** | ✓ | ✓ | ✓ | ✓ | ✓ | ✓ | ✓ |

Table 1: Comparison of our survey with related works, organized by research type and focus (DP: data preparation)

Table 1 does not constitute a qualitative assessment of these research works, but rather highlights the differences in scope and intent between them. Nonetheless, in most cases, our

survey does tackle data preparation facets in a more extensive and in-depth manner than similar works, in terms of data preparation components and analyzed systems and methods.

## 3 Survey Methodology

The primary motivator for this survey is the gap in automated data science and machine learning research when it comes to data preparation. The literature on automated data preparation is scarce, which led us to conduct an in-depth study of the topic.

A starting point was the *Automating Data Science* (De Bie et al., 2022) paper, which outlines the prospects and challenges of automating data science, including data preparation. In order to perform a thorough survey, we mainly relied on broad paper search engines with different keyword combinations, and backward snowballing (examining the references cited in papers).

The literature we surveyed, covering multiple domains, predominantly falls into following categories (with primary related keywords):

1. *AutoML surveys, benchmarks, systems*—to examine the state of data preparation in AutoML. Keywords: *automl, automated machine learning, survey, benchmark.*

2. *Data-centric AI surveys and position papers*—to establish the value of data quality and optimizing data. Keywords: *data-centric, artificial intelligence, machine learning, survey.*

3. *Data transformations: books, surveys, and methods*—to understand the complete scope of data preparation. Keywords: *data transformation/ preparation/ wrangling/ integration/ cleaning/ preprocessing, feature engineering, [individual methods].*

4. *Impact and evaluation*—papers that help understand and quantify the impact of data preparation in (Auto)ML, and how to evaluate it. Keywords: *data preparation/ preprocessing, quality, impact, effect, measuring, evaluation, benchmark.*

5. *(Semi-)automated data preparation systems*—to examine existing automated data preparation solutions. Keywords: *automated, data science/ preparation/ wrangling/ integration/ cleaning/ preprocessing, feature engineering.*

6. *Optimization methods*—to better understand underlying methods. Keywords: *[individual methods].*

Papers were included according to their relevance to automated data preparation and the related areas listed above, particularly with regard to their pertinence in helping to accurately define and describe concepts or processes, and answer various questions or concerns. We prioritized works published in reputable peer-reviewed journals and conferences, with a few exceptions for work from other sources, such as books, well-established systems and methods, or recent preprints. The selection processes for AutoML and (semi-)automated data preparation systems are explained in more detail in their dedicated sections.

# 4 Data Preparation: Transformations

In the context of machine learning, data preparation transformations are operations applied to data, with the end goal of enhancing results when combined with a model (Aggarwal, 2015; García et al., 2015; Sakr and Zomaya, 2019; Brownlee, 2020; Hameed and Naumann, 2020; Fernandes et al., 2023). The pool of possible transformations is large, and the relevance of each transformation depends on the data itself, but potentially also the ML task to be performed, and the model employed for the task. To facilitate navigation among these possibilities, we extract a structure based on the aspects of data preparation that each transformation addresses.

## 4.1 Taxonomy Structure

We propose a taxonomy of data preparation transformations with three levels of categorization: transformation (1) purpose, (2) function, and (3) category. A visual representation of this structure is depicted in Figure 2.

On the highest level, transformations can be characterized by what they aim to achieve, i.e., their *purposes*: data organization, data quality, and model performance (Nazabal et al., 2020). Rather than always having precise cutoffs, transformation purposes form a spectrum such that transformations can serve multiple purposes to varying degrees. For example, aligning data contents is an organizational task that also results in higher-quality data, which eventually translates to better model predictions. In the interest of clear structuring, we attribute transformations to primary purposes. We also note that each purpose is generally conducive to the next one: good organization is reflected in data quality, and quality tends to boost the results achieved by models.

We further differentiate transformations by their *functions*, the functional facets through which they advance their purposes: data integration, data cleaning, data preprocessing, and feature engineering. Similarly to purposes, there is some overlap between adjacent transformation functions.

Finally, we assign transformations to multiple *categories*, by grouping together different transformation methods that address a common data preparation element—whether resolving a common problem with the data (e.g., different ways to handle missing values), or improving a common aspect of the data (e.g., different ways to generate data points).

Below are descriptions of data preparation purposes, functions, and the categories of transformations enveloped within, along with transformation method examples of each type. We note that this is not a complete catalog of all possible data transformations, particularly if we consider the prospect of creating custom operations optimized for each individual dataset. In practice, transformation ordering often tends to align with the presented order, however this is not a mandatory pattern.

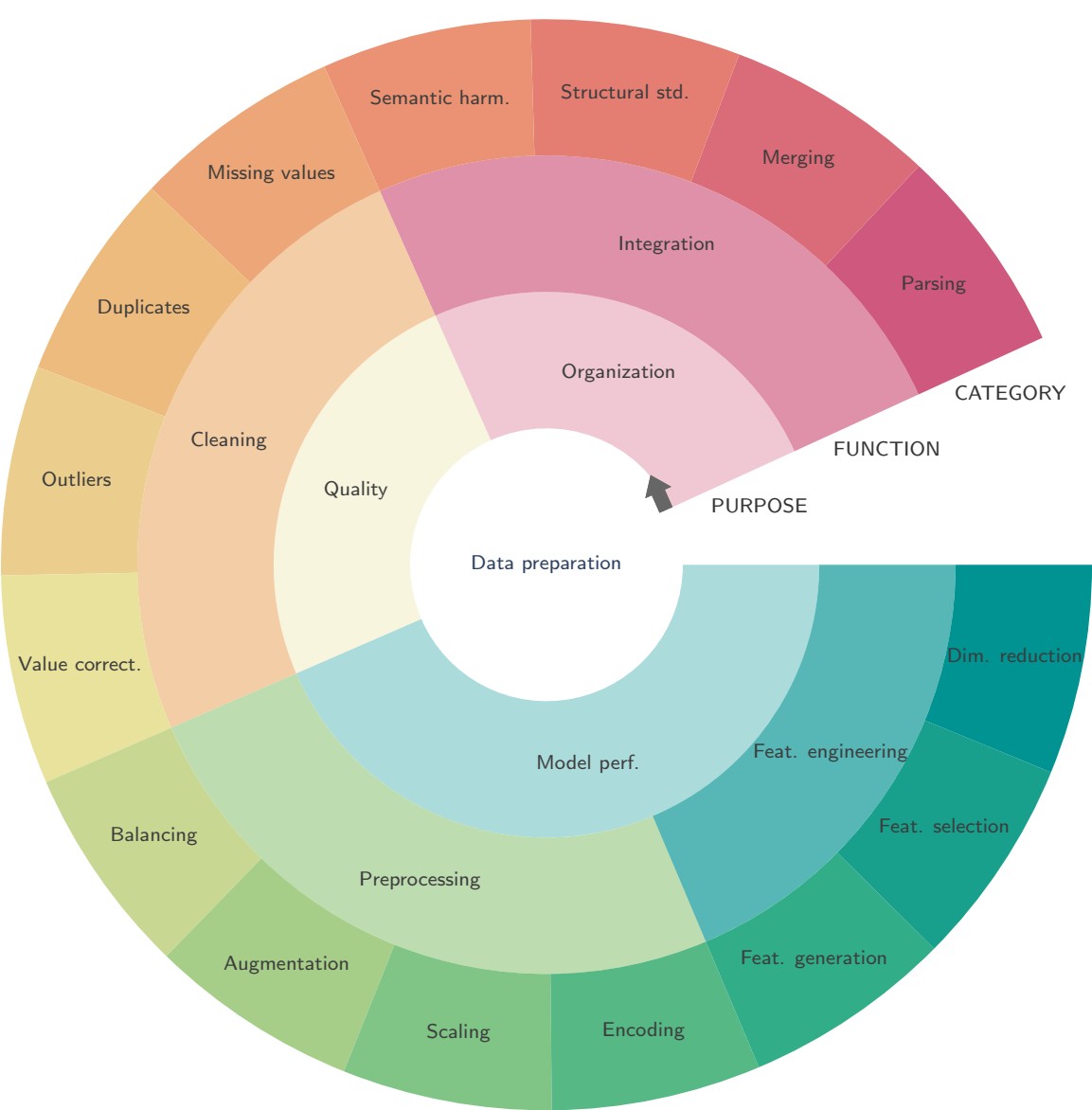

Figure 2: Taxonomy of data preparation transformations

## 4.2 Data Organization

Organizing data implies shaping it into a format that is well-suited for analytics and further exploitation. For tabular data, this is commonly a single dataset with data points as rows and features as columns. In the case of supervised learning, a target variable column is also included. Other data modalities can often be represented in tabular format, e.g., time series data with one of the features consisting of temporal values so that each sample is associated with a timestamp, images formatted as rows of pixel values, or text data with extracted characteristics as features. They may also have a different native format better adapted to their features.

### 4.2.1 DATA INTEGRATION

Data integration (Lenzerini, 2002; Ziegler and Dittrich, 2007; Doan, 2012), also known as ETL (Extract, Transform, Load), consists in putting together data from different sources and in different formats, to constitute a single coherent dataset, in a format that can be input into an ML model and processed by it. Integration concerns both outward formatting and content organization.

*Parsing* refers to extracting data from a variety of sources and formats and loading their contents into structured datasets.

> *Examples.* Importing data from databases, spreadsheets, or files in varied formats (e.g., images, text documents) into datasets.

*Merging* designates combining the contents of multiple datasets, with possibly different structures, into a single dataset.

> *Examples.* Vertical, horizontal, or diagonal concatenation. Inner, outer, left, or right joins.

*Structural standardization* focuses on consistent and usable data representation in terms of data format and types, reconciling inconsistencies, and ensuring support and interpretability by data preprocessing operations and ML models.

> *Examples.* Type conversion: casting booleans to categorical or numeric types; casting complex types to more interpretable ones (e.g., dates in string format to date/time or numeric formats); simplifying composite types (e.g., exploding lists of values into separate data points). Unit standardization (e.g., converting *km*, *m*, *cm* to meters). Schema alignment or matching (Rahm and Bernstein, 2001) (e.g., unifying "birth_date" and "DOB" features). Image format manipulations (e.g., converting to a common format and color space; resizing or cropping to a common shape).

*Semantic harmonization* is concerned with the meaning of the data, handling cases where the same information content appears in multiple datasets, features, or data points, but with inconsistent terminology or form.

*Examples.* Entity resolution (Getoor and Machanavajjhala, 2012; Christophides et al., 2020; Zhao et al., 2020) (e.g., recognizing "John Smith", "J. Smith", "Smith, John" as the same entity), for instance via record linkage (Winkler, 2014), i.e., linking entities based on shared attributes; or fuzzy matching (Navarro, 2001), i.e., approximate string matching. Value mapping (e.g., mapping synonyms to a common value). Semantic, or column, annotation (Uren et al., 2006): associating additional meaning with data contents (e.g., noting that a "DOB" feature designates date of birth and contains temporal values).

## 4.3 Data Quality

High-quality data is crucial for trustworthy analysis, and provides a strong foundation for inference (Mohammed et al., 2025). Independent of the chosen model, data quality can be improved by making sure the data is complete, consistent (no conflicting information), representative, and interpretable. Data quality typically sets an upper bound on model performance with regard to reality (Li et al., 2021; Whang et al., 2023).

### 4.3.1 Data Cleaning

Data cleaning (Rahm and Do, 2000; Dasu, 2004; Chu et al., 2016) targets imperfections in the data contents, such as errors, inconsistencies, or the presence of unnecessary elements; ensuring the data can be used by a model, and raising its general quality by repairing or removing these imperfections.

*Missing values* refer to completing data by detecting and handling null values in the dataset.

> *Examples.* Removing missing values. Imputing (Lin and Tsai, 2019), or filling, them using different methods (e.g., forward/backward filling, i.e., using the value of the next or previous data point; interpolation; the $k$-nearest neighbor algorithm) (Peterson, 2009). Encoding "missing-ness" as a new feature.

*Duplicates* designate detecting and handling duplicated data points in a dataset, so as to avoid incorrect biases.

> *Examples.* Removing duplicates. Encoding them as a new feature representing the number of appearances of each data point.

*Outliers* address detecting and possibly removing outliers, i.e., data points that significantly differ from the rest of the data or their surroundings, and could represent errors or noise.

> *Examples.* Detection using standard deviation or IQR (interquartile range) (Hodge and Austin, 2004).

*Value correction* deals with detecting and repairing incorrect or inconsistent values in the data. It concerns feature value repair (Bertossi, 2019), such as spelling errors, out-of-range measurements, or examples such as a mismatched city and postal code (often referred to as record repair in a database context); as well as label correction (Frenay

and Verleysen, 2014; Song et al., 2023) when the error concerns the target variable, i.e., wrongly labeled data points.

*Examples.* Rules and constraints (e.g., age must be $\geq 0$). Outlier value detection (c.f. *Outliers* above) combined with statistical (e.g., mean, mode) or ML-based (e.g., classification, regression) imputation.

## 4.4 Model Performance

Model performance optimization refers to transforming data in order to improve the outcomes of a particular ML model. The main performance criterion is usually a chosen effectiveness metric for that model, but there can also be considerations for efficiency, prompting additional optimization for execution time and resource usage (Nawi et al., 2013; Motamedi et al., 2021).

### 4.4.1 DATA PREPROCESSING

Data preprocessing addresses the distribution and contents of data points in order to make the data more robust and facilitate correct interpretation by a model, raising its performance. Preprocessing makes it possible to leverage the superior efficiency of many ML models when it comes to working with numeric data. Moreover, changing the number of data points can, in addition to balancing out data distribution, also affect time and memory usage due to differences in the amount of data to pass through the model.

*Balancing* mitigates issues relating to imbalances in data distribution, typically for classification tasks, via resampling (Carvalho et al., 2025; Khushi et al., 2021; Bagui and Li, 2021): oversampling or undersampling, that is, strategically adding data points to underrepresented classes (i.e., minority classes), or removing data points from overrepresented classes (i.e., majority classes).

> *Examples.* Oversampling methods—random oversampling: duplicating random data points in minority classes; SMOTE (Synthetic Minority Oversampling Technique) (Chawla et al., 2002): generating new points via interpolation between minority class data points and their nearest neighbors; ADASYN (Adaptive Synthetic algorithm) (He et al., 2008): an extension of SMOTE specifically targeting the lowest density areas of minority class data points. Undersampling methods—random undersampling: removing random data points in majority classes; ENN (Edited Nearest Neighbors) (Wilson, 1972): removing majority class points located in a neighborhood of predominantly minority class points; Tomek's Links (Tomek, 1976): removing majority class points whose nearest neighbor is a minority class point.

*Data augmentation* (Mumuni and Mumuni, 2022) consists in the synthetic generation of new data points, by altering or combining feature values from existing ones.

> *Examples.* Augmentation methods can coincide with resampling ones that generate synthetic data points, such as SMOTE and ADASYN mentioned above. There can be

more diverse transformations for some data modalities (e.g., for images: flipping, rotating, cropping; or for text: replacing words with synonyms, selective word dropout).

*Scaling* designates the scaling of numeric features by adjusting their range or distribution, while preserving the ordering between values. It brings consistency, ensuring that no feature dominates others due to its magnitude. Feature scaling enhances performance for many kinds of models, and is particularly useful when feature values are interpreted as distances between data points, such as in nearest-neighbor algorithms.

*Examples.* Normalization: proportionally scaling to a fixed range, usually [0, 1]. Standardization: centering data so that the mean becomes 0 and the standard deviation 1. Robust scaling: similar to standardization, but using the mean and interquartile range (de Amorim et al., 2023).

*Encoding* (Chatfield et al., 2011) stands for the encoding of features (e.g., categorical features in the form of strings) to numeric representations (scalars or vectors), so that they can be used by models accepting only numeric features, which are quite common.

*Examples.* Hashing: applying a hash function to create a mapping between categorical and numeric values. One-hot encoding: converting features to a set of binary indicator features corresponding to each of the original features' possible values (where, for a given observation, the value 1 indicates the original feature value, and the remaining values are 0) (Kozina et al., 2024).

### 4.4.2 FEATURE ENGINEERING

Feature engineering corresponds to the manipulation of features in order to optimize model performance. Curating the selection of features and the information within them can help avoid overfitting the model and improve time performance, while mostly preserving data quality and information. It is also possible to derive added information from existing features, improving model results.

*Feature generation* focuses on generating new features by extracting or transforming information from existing ones.

*Examples.* Applying numeric operations (e.g., logarithm, square root). Discretization (Dougherty et al., 1995): transforming continuous features to discrete ones, often with binning techniques (e.g., uniform, quantile, $k$-means clustering) (MacQueen, 1967). Extracting temporal data (e.g., the day of the week from a date).

*Feature selection* refers to reducing the number of features by making a selection of those most relevant to the target variable.

*Examples.* Forward feature selection: incrementally selecting one feature at a time. Coefficient of variation (i.e., ratio of variance to mean) threshold. RFE (Recursive Feature Elimination) decision trees. Univariate correlation to the target (Li et al., 2017).

*Dimensionality reduction* (van der Maaten et al., 2009; Sorzano et al., 2014; Jia et al., 2022) means reducing the number of features through transforming the data into a lower-dimensional representation, by combining features while retaining essential information.

*Examples.* PCA (Principal Component Analysis) (Pearson, 1901; Abdi and Williams, 2010): a linear technique of projection to a coordinate system where the greatest variance lies along the first axis or principal component, the second greatest variance along the second axis, and so on. *t*-SNE (*t*-distributed Stochastic Neighbor Embedding) (van der Maaten and Hinton, 2008): a non-linear technique preserving local relationships between data points. UMAP (Uniform Manifold Approximation and Projection) (McInnes et al., 2018): a non-linear technique preserving both local and global structures.

### 4.5 Data Modality Specifics

The transformation categories described above represent a broad overview of data preparation elements that generally support transformations for multiple data modalities. Our examples often concern tabular data, but also incorporate instances of other modalities. Beyond our taxonomy and examples, there also exist modality-specific preprocessing transformations, which could be considered in a wider context. Examples include image manipulation such as filtering or segmentation (Sonka et al., 1993), or NLP (natural language processing) techniques for textual data, such as tokenization, vectorization, or translation methods (Patil et al., 2023).

### 4.6 Data Usability

One additional concept to consider is that of data *usability*. In the context of machine learning, data being usable means that it can be processed by a given ML model. Usability criteria can differ for different models. For instance, some models cannot run on incomplete data, while others can handle missing values; some models can only process numeric data, meanwhile others support a wider variety of types. While not a goal in itself, usability is a requirement for an ML system to function. It is achieved by applying data transformations that address specific incompatibilities—in the previous examples, handling missing values, encoding certain data types, or scaling numeric values.

### 4.7 Caveats

While transformations ensuring base compatibility between dataset and model may be indispensable, the contributions of others can be more situational. The varying effectiveness of data transformations in different cases has been pointed out in analyses of some specific transformations. Examples include a study on the diminishing returns of missing value imputation (Morvan and Varoquaux, 2025), or another that shows that balancing data with the SMOTE algorithm is more pertinent for weaker classification models, and less so in the opposite case (Elor and Averbuch-Elor, 2022).

Another aspect to consider when selecting transformations is their potential effect on information content (Knobbout et al., 2019). Information loss most commonly stems from the removal of parts of the data, such as missing out on some fringe cases when removing outliers, or slightly degraded predictions when omitting less relevant features. Altering information content can have deeper effects on data analytics and ML result interpretation, as exemplified in a fairness study (Biswas and Rajan, 2021) showing that certain data preparation elements can introduce or enhance biases in the data, and suggesting alternatives to mitigate this effect.

Additionally, explainability can also be affected by data preparation (Gonzalez Zelaya, 2019; Strasser and Klettke, 2024). Explainability is tightly connected to the transparency of selected data transformations. It can be enhanced by some integration and cleaning operations, but also reduced by non-invertible transformations that render the data more obscure, such as the PCA algorithm or various aggregation functions. Result explainability also depends on the selected ML model (Rudin, 2019).

## 5 Data Preparation: Pipelines

The data preparation process consists in constructing a pipeline of transformations to be applied to the data, as part of the larger machine learning pipeline wherein it accompanies modeling. In this section, we describe how data pipelines are designed and evaluated.

### 5.1 Pipeline Design

The design of data preparation pipelines is aimed at elevating the quality of the data, in addition to ensuring compatibility and optimizing synergy with a model, all in service of enhancing results. These goals can be achieved by optimizing a data transformation sequence on datasets adequately set up for the task.

#### 5.1.1 Downstream Task Importance

Some experiments have suggested that, for certain datasets, optimal pipeline design can be model-independent (Quemy, 2019). Nevertheless, there is no single best data preparation pipeline that is the same for every possible problem, nor can we truly affirm that one pipeline is *a priori* better than another in isolation. This question has notably been addressed for data integration (Soong and de Montigny, 2004), as well as for learning on imbalanced data (Moniz and Monteiro, 2021). It has also been studied for entire pipelines (Giovanelli et al., 2022).

In fact, not every data quality aspect or data transformation necessarily influences the performance of every model in the same way. For example, in classification over unevenly distributed data, a neural network classifier can significantly improve after resampling the data to balance out its distribution (Carvalho et al., 2025), while random forest models tend to be more robust and may not need this kind of preprocessing to achieve similar results (Khoshgoftaar et al., 2007; Dittman et al., 2015).

This highlights the importance of not only the data format and contents, but also the machine learning task to perform, and the model selected for the task, in the pipeline design process. Thus, it is crucial consider both the data and the downstream task when optimizing data pipelines.

### 5.1.2 TRAINING AND EVALUATION CONSIDERATIONS

As with the modeling process, where an ML model is trained on a training dataset and evaluated on a test dataset, data pipeline optimization is also done with the help of the training set. The same sequence of transformations is later applied to the test set before evaluating the model on it, with a few exceptions—transformations that risk skewing the evaluation process, such as balancing transformations, which can distort the data distribution by adding or removing data points, are not transferred to the test set. Additionally, the inclusion of some transformations in the pipeline requires that their inverses be applied before evaluation. For example, if the target variable is encoded during preprocessing, comparisons with (unencoded) ground truth will fail—the variable therefore needs to be decoded. The process for any other non-training data (such as a validation set or newly incoming data) is the same as that of the test set.

Another point of importance with regard to the train-test mechanism is the process of separating a dataset into training and test data. In a controlled environment, the two sets should be disjoint, however in real-world conditions some overlap can occur. In order to enable robust learning, so that patterns learned on the training set can be relevant to a previously unseen test set, the two datasets should be sampled from the same underlying data distribution in a representative way. Despite the most frequent dataset splitting method being the random split—randomly assigning data points to the two sets while only considering their proportional sizes—distribution similarity between datasets can be promoted through the use of strategic sampling methods, such as $k$-fold cross-validation (Stone, 1974; Refaeilzadeh et al., 2009). A well-stratified and robust data split increases the odds of finding an optimal data preparation pipeline in terms of generalization.

In addition to stratified sampling, ML processes often implement strategies to mitigate the phenomenon of overfitting (Dietterich, 1995). In the context of data pipelines, overfitting occurs if the pipeline design or its inner operations fit too closely to training data specifics, for instance capturing noise, random fluctuations, or artifacts—which leads to poor generalization to new, unseen data. Therefore, a common ML practice is to split the data into three datasets: (1) the training set is used to fit the pipeline (e.g., hyperparameters), (2) the validation set is used to provide feedback to the training process with regard to pipeline generalization performance (e.g., using AutoML), and, since repeated evaluation on the validation data can also lead to overfitting of the pipeline design, (3) the test set is used on the final optimized data pipeline to obtain a real, unbiased, estimate of its generalization capabilities to new data. For both levels of splits (i.e., between training, validation, and test), different approaches can be employed, such as holdout splits, cross-validation, or other (nested) resampling strategies, typically depending on the chance of overfitting and the cost of doing evaluations.

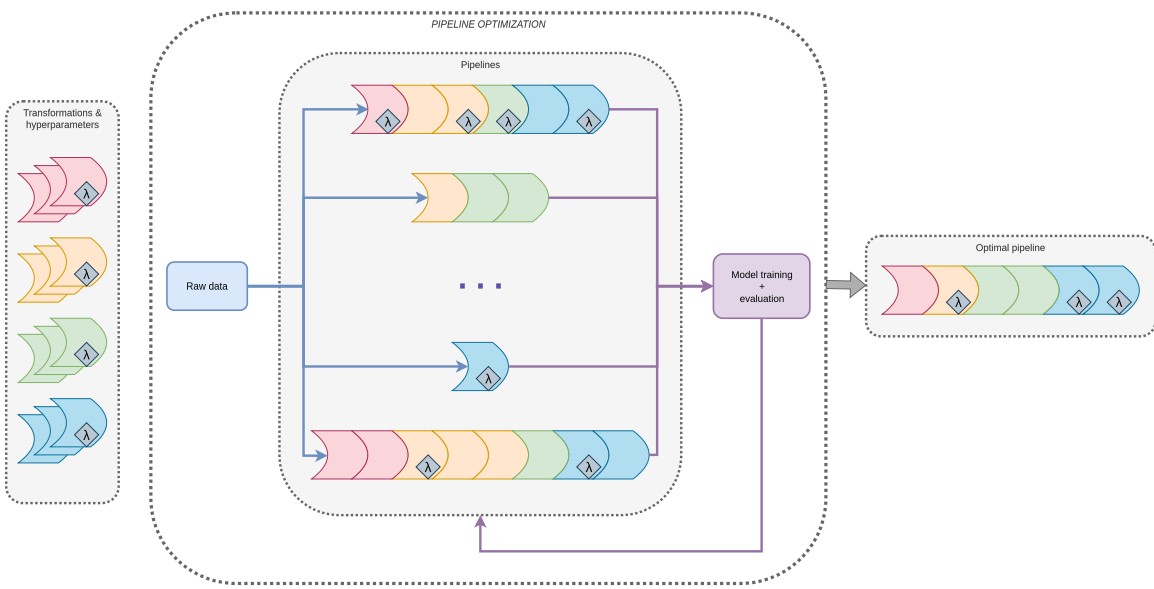

Figure 3: Data preparation pipeline optimization

### 5.1.3 OPTIMIZATION PROBLEM

For a given ML algorithm, the main considerations in optimizing a data preparation pipeline are:

1. *Transformation selection*: choosing data transformations to apply and sometimes which features to apply them to;

2. *Transformation ordering*: architecturing the sequence of transformations;

3. *Hyperparameter optimization*: configuring transformation hyperparameter values where applicable.

The pipeline optimization process is illustrated in Figure 3. Handling the three aforementioned objectives allows for much flexibility. A data pipeline can contain multiple instances of the same data transformation, possibly applied to different sets of features and with different hyperparameter configurations. Though data pipelines are commonly structured as linear sequences of transformations (as depicted in the figure), they can also adopt acyclic graph structures (Olson and Moore, 2016), where some transformations are executed in parallel. In this case, we can talk of a sequence of transformation steps, where each step can consist of one or more simultaneous transformations.

Since the impact of data pipeline choices cannot be assessed without empirical evaluation, we can regard pipeline design as a *black-box optimization problem* (Audet and Hare, 2017) whose search space lies in the possible combinations along the three enumerated dimensions.

We can formally define the problem as follows. Let:

- $D$ be a dataset, and $D_{\text{train}}$, $D_{\text{val}}$ and $D_{\text{test}}$ its subsets used for training, validation, and testing respectively;

- $A$ be a machine learning algorithm with a fixed hyperparameter configuration;

- $\mathcal{T}$ be the set of all possible data transformations, with $T \in \mathcal{T}$ a transformation instance;

- $\Lambda_T$ be the hyperparameter configuration space of a transformation $T$, with $\lambda_T \in \Lambda_T$ one specific hyperparameter configuration of $T$;

- $\mathcal{P}$ be a data preparation pipeline configuration space, with elements $P \in \mathcal{P}$. In a linear pipeline structure, elements are in the form of $P = \big((T_1, \lambda_{T_1}), (T_2, \lambda_{T_2}), ..., (T_n, \lambda_{T_n})\big)$, $n \in \mathbb{N}$. For every base pipeline $P$ used for training, there is a derived pipeline $P'$ adapted for evaluation (which may be identical or slightly differ from $P$, as described in Section 5.1.2);

- $\mathcal{M}$ be an evaluation metric for predictive performance, with higher values indicating better performance.

The optimal solution to our data pipeline optimization problem lies in solving Equation (1):

$$P^* \in \arg\max_{P \in \mathcal{P}} \ \mathcal{M}\Big(A\big(P(D_{\text{train}}), \ P'(D_{\text{val}})\big), \ P'(D_{\text{test}})\Big) \tag{1}$$

i.e., maximizing the result achieved by a model trained with algorithm $A$ using datasets $D_{\text{train}}$ transformed by pipeline $P$ and $D_{\text{val}}$ transformed by pipeline $P'$, when evaluated according to metric $\mathcal{M}$ on dataset $D_{\text{test}}$ transformed by pipeline $P'$.

For brevity, we can denote Equation (1) as:

$$P^* \in \arg\max_{P \in \mathcal{P}} \ \mathcal{M}(D, \ P, \ A) \tag{1'}$$

Data pipeline optimization can also be considered in combination with the *CASH* (Combined Algorithm Selection and Hyperparameter optimization) (Thornton et al., 2013) problem, which is central to AutoML. The search space is then extended to combinations of data pipelines, ML algorithms, and algorithm hyperparameters. In addition to the previous problem setup, let:

- $\bar{\mathcal{A}}$ be the set of possible ML algorithms, with $\bar{A} \in \bar{\mathcal{A}}$ one (unconfigured) algorithm;

- $\Lambda_{\bar{A}}$ be the hyperparameter configuration space of algorithm $\bar{A}$, with $\lambda_{\bar{A}} \in \Lambda_{\bar{A}}$ one specific hyperparameter configuration of $\bar{A}$;

- $\mathcal{A}$ be the set of possible ML algorithms and implicitly their possible configurations, where each element $A \in \mathcal{A}$ has the form of $A = (\bar{A}, \lambda_{\bar{A}})$.

To include ML algorithm selection and associated hyperparameter optimization, Equation (1$'$) is expanded into Equation (2):

$$(P,\ A)^* \in \arg \max_{A \in \mathcal{A},\ P \in \mathcal{P}} \ \mathcal{M}(D,\ P,\ A) \qquad (2)$$

### 5.1.4 PROBLEM VARIANTS

Often, particularly in real applications, data pipeline optimization is subject to additional considerations. These can be represented as constraints or additional objectives. Common variants of the problem include limited resource availability, such as limited time or processing power, which can translate to optimization under constraints, e.g., a restricted evaluation budget, or into optimizing a multi-objective function. Another common variant is one requiring explainability. The introduction of such considerations can affect the contents of the search space, as well as the optimal way of exploring it.

### 5.1.5 EXPERIENCE AND DOMAIN INSIGHTS

Beyond exploring the data pipeline search space, data preparation can be facilitated by learning from experience with different datasets and pipelines, as well as by leveraging domain knowledge, which implies knowledge about the data's meaning, structure, and context. When available, these two kinds of insights allow for informed decision-making rather than operating in a purely black-box environment, leading to more efficient search space navigation and faster convergence towards an optimal solution.

## 5.2 Pipeline Evaluation

Evaluating data preparation pipelines in a meaningful and objective way is quite challenging (Kumar, 2021; Aroyo et al., 2022). As of yet, there are no standard benchmarks, methods, or metrics specific to it. Instead, impact is usually quantified indirectly, through assessments of data quality and model performance.

### 5.2.1 EVALUATION THROUGH DATA QUALITY

Data quality is of paramount importance when it comes to achieving reliable ML results that adequately represent reality (Mohammed et al., 2025; Li et al., 2021; Whang et al., 2023). Evaluating data pipelines through the lens of data quality implies comparing data quality before and after the application of a data preparation pipeline. However, defining and measuring data quality is in itself a topic of discussion (Pipino et al., 2002; Batini et al., 2009; Lawrence, 2017; Jain et al., 2020; Seedat et al., 2022). Identifying quality facets to consider and ways in which to quantify them are non-trivial problems.

In addition, data quality issues can be domain-specific (Foidl et al., 2022), which adds a layer of complexity to the treatment of such data. For instance, in the case of outliers, without domain knowledge it can be hard to tell whether an outlying point is an error, or actually constitutes a relevant data point.

Moreover, it is worth recalling the importance of the ML model in data pipeline optimization (cf. Section 5.1.1). Consequently, this is also a major factor in evaluation. Evaluating pipelines purely based on data quality runs the risk of an inaccurate assessment, due to not capturing their performance with regard to the downstream task.

### 5.2.2 Evaluation through Model Performance

The most common way to evaluate data preparation is through downstream model performance, i.e., assessing the model according to a chosen metric. While this is certainly a pertinent indicator, it is in fact a measure of how well the model fits the data, rather than how well the model fits reality. Equating strong model performance with good data preparation can be risky due to questions of data quality and the possibility of underlying biases.

In order for results to be pertinent, it is also particularly important for the model evaluation metric to be carefully selected so as to be adequate for the problem. Even then, model performance may not be entirely reliable, since its score can increase from overfitting. In actuality, this makes the model worse, though the effect can be mitigated through validation strategies, e.g., cross-validation (Stone, 1974; Refaeilzadeh et al., 2009), or robustness techniques such as early stopping or regularization (Srivastava et al., 2014; Ying, 2019).

This method of evaluating data preparation by measuring model performance is therefore directly dependent on data quality and the robustness of the model evaluation. It is also quite costly, since it requires a model to be trained on every pipeline. However, this remains the most accessible and dependable evaluation method to date.

### 5.2.3 Transformation Interactions

If we consider individual or groups of transformations within a pipeline, evaluation is additionally complicated by interactions between data transformations: applying one transformation can affect the behavior of a subsequent one. For example, removing outliers can change the span of values of a feature, and consequently the effectiveness of normalization. This makes it very difficult to measure the impact of a single transformation in isolation, or to identify which parts of the pipeline contribute to performance improvement or degradation, though there are some promising beginnings in this direction (Gonzalez Zelaya, 2019; Strasser and Klettke, 2024).

### 5.2.4 Benchmarks, Tools, and Other Evaluation Methods

Setting aside methodology, we also encounter a lack of standardized benchmarks for data pipeline evaluation (Oala et al., 2024). Unlike certain ML subfields that benefit from well-known datasets conventionally used for their model-centric benchmarks, such as *ImageNet* (Deng et al., 2009), *CIFAR-10/100* (LeCun et al., 1998) and *MNIST* (Krizhevsky, 2009) for image classification, or *GLUE* (Wang et al., 2019) for natural language processing, the domain of data preparation, or more broadly that of data-centric ML, has yet to establish a consensus on benchmarking criteria and datasets (Oala et al., 2024).

Nonetheless, some recent works have begun to tackle various aspects of evaluating data preparation pipelines specifically. One proposition is an automated *provenance-based screening* (Schelter et al., 2024) method, modeling data preparation pipelines into *dataflows* used to detect certain types of mistakes and generate metadata information. The *ML Data Prep Zoo* (Shah and Kumar, 2019) presents a repository of data preparation tasks, labeled benchmark datasets, and pre-trained ML models, providing the tools to create and evaluate automated solutions to various data preparation tasks. The GOUDA (Restat et al., 2022) tool automatically generates flawed datasets that, along with their ground truth, facilitate the analysis and evaluation of data preparation pipelines. DCBENCH (Eyuboglu et al., 2022) is a benchmark for the evaluation of select components of data-centric AI systems. DATA-PERF (Mazumder et al., 2023) provides another collection of data-centric algorithm benchmarks for multiple ML tasks, as well as a community platform allowing new benchmarks to be added. Additional benchmarking datasets can be found on the *OpenML* (Vanschoren et al., 2014; Bischl et al., 2025) platform. Such methods, tools, and collections pave the way towards a better understanding and a more structured approach to pipeline design and evaluation.

## 6 Automated Data Preparation in AutoML

In order to assess the state of data preparation in automated machine learning (AutoML), we explore the data transformation categories implemented by different AutoML solutions, as well as the decision processes in charge of handling them.

### 6.1 Overview of Relevant AutoML Systems

We compile a collection of AutoML systems relevant to this survey according to the following criteria:

*(1) Providing end-to-end automation.* This excludes approaches that are not fully automated and require human interaction. Notable systems eliminated by this criterion include: ALPINE MEADOW (Kraska, 2018; Shang et al., 2019), AUTOML-DSGE (Assunção et al., 2020), AUTOSTACKER (Chen et al., 2018), AUTO_VIML (featurewiz, 2019), ATM (Auto Tune Models) (Swearingen et al., 2017), DABL (dabl, 2016), LUDWIG (Molino et al., 2019), MOSAIC (Rakotoarison et al., 2019), NNI (Neural Network Intelligence) (Microsoft, 2018), RECIPE (REsilient ClassifIcation Pipeline Evolution) (de Sá et al., 2017), TRANSMOGRI-FAI (Salesforce, 2017).

*(2) Covering the whole ML pipeline or at minimum the whole modeling phase.* This excludes approaches addressing only inner ML components or parts of the pipeline that do not interact with data preparation directly, e.g., neural network search (NAS) systems such as NASLIB (Ruchte et al., 2020), KATIB (Zhou et al., 2019), HYPERNETS (Yang et al., 2020b).

*(3) Being open source and providing public documentation with enough information for our study.* This ensures that our information about the considered systems is reliable and relatively easily verifiable. This selection notably excludes closed-source commercial systems from some of the biggest industry players in AI and AutoML: *Google Cloud AutoML* (Google,

2018), *DataRobot AI Cloud* (DataRobot, 2015), *H2O.ai Driverless AI* (H2O, 2020), *Microsoft Azure Machine Learning AutoML* (Microsoft, 2018), *Amazon SageMaker Autopilot* (Amazon, 2019), *Oracle AutoML* (Yakovlev et al., 2020). It is, however, worth noting that these typically do contain data preparation elements. Published research solutions lacking supporting code, such as ADMM (Alternating Direction Method of Multipliers) (Liu et al., 2020) AutoML and AutoCompete (Thakur and Krohn-Grimberghe, 2015), are also excluded.

*(4) Being reasonably usable (projects that are still maintained, or left in a stable state).* This eliminates deprecated, outdated or unmaintained solutions.

Below are the AutoML systems that we retain for consideration. We categorize them according to the way in which they handle data preparation: separating those that perform preset or rule-based data treatment—systems with static data preparation, and those whose optimization algorithm also covers data preparation pipelines—systems with optimized data pipelines. We do not differentiate systems by the ML tasks they target or other modeling factors, keeping our focus on data preparation aspects instead. The majority of these systems work with tabular data, while several also offer support for additional data modalities, which we highlight in their descriptions.

### 6.1.1 Static Data Preparation Systems

Oboe (& TensorOboe) (Yang et al., 2019, 2020a) is a meta-learning-based (Schmidhuber, 1987; Hospedales et al., 2021) AutoML system using collaborative filtering (Goldberg et al., 1992; Sarwar et al., 2001) to predict model performance under resource constraints; TensorOboe extends the method with tensor completion (Liu et al., 2013) and adds more robustness with regard to data quality.

AutoML-Zero (Real et al., 2020) is a research prototype exploring the evolution (Bartz-Beielstein et al., 2014) of complete machine learning algorithms from a set of basic mathematical operations, i.e., "from zero".

FLAML (Fast and Lightweight AutoML) (Wang et al., 2021) is an AutoML library designed for efficiency, using lightweight learners and resource-aware tuning without heavy search. It supports integration with the *Fabric* data platform, as well as the MLOps platform *MLflow*.

AutoKeras (Jin et al., 2023) is an AutoML system based on the Keras (Chollet, 2015) deep learning library, which performs neural architecture search and model tuning on image, text, and multimodal data. As of today, tabular data is no longer included in the official module, though it is available through an extension (De Bruin, 2017).

H2O AutoML (LeDell and Poirier, 2020) is an enterprise-ready, scalable AutoML solution using ensembling (Dietterich, 2000), and offering model interpretability and integration with the *H2O* analytics platform.

BLUECAST (ThomasMeissnerDS, 2023) is a fast and lightweight AutoML library mainly featuring XGBoost (Chen and Guestrin, 2016) models. It also includes optional user customization and a data toolkit for more advanced tasks, as well as explainability features.

AUTO_ML (Parry, 2016) is a simplified AutoML package designed for production. It allows automatic training of multiple models of the same kind based on a chosen data split (e.g., a separate model for every country from worldwide data).

MLJAR-SUPERVISED (MLJAR, 2018) is an AutoML tool with different modes for fast prototyping, explainability, and deployment-ready outputs.

TABPFN (Hollmann et al., 2022, 2025) is a foundation model (Bommasani et al., 2022) for small tabular datasets that near-instantly produces highly accurate predictions, without the need for hyperparameter tuning.

AUTOGLUON (Erickson et al., 2020; Shchur et al., 2023; Tang et al., 2024) is a versatile AutoML solution supporting tabular (AUTOGLUON-TABULAR), time series (AUTOGLUON-TIMESERIES), and multimodal (AUTOMM) data with images and text. It performs automatic model selection, tuning and ensembling.

NAÏVE AUTOML (Mohr and Wever, 2022) is a naïve AutoML tool useful for baseline comparisons, that combines basic preprocessing with random search. Its efficiency lies in optimizing each pipeline component in isolation, thus significantly reducing its search space.

MLBOX (de Romblay, 2017) is an AutoML library focused on data drift detection (Ackerman et al., 2020), preprocessing, and model optimization.

LIGHTAUTOML (Vakhrushev et al., 2022) is an AutoML library optimized for performance, interpretability, and production deployment.

ML.NET AUTOML (Ahmed et al., 2019) is a framework for .NET developers, which provides seamless integration with .NET applications. It supports tabular, text, and image data.

### 6.1.2 OPTIMIZED DATA PIPELINE SYSTEMS

AUTO-WEKA (& 2.0) (Thornton et al., 2013; Kotthoff et al., 2017) is an extension of the WEKA (Holmes et al., 1994) data platform that automates algorithm and hyperparameter selection using Bayesian optimization (Snoek et al., 2012; Garnett, 2022); v2.0 features multi-objective search.

HYPEROPT-SKLEARN (Komer et al., 2014) is a wrapper for scikit-learn (Pedregosa et al., 2011) models with Hyperopt-based (Bergstra et al., 2013) Bayesian optimization for algorithm and hyperparameter optimization.

TPOT (The Tree-based Pipeline Optimisation Tool) (Olson and Moore, 2016; Gijsbers et al., 2017; Parmentier et al., 2019) is an AutoML system that uses genetic programming (Bartz-Beielstein et al., 2014) to evolve ML pipelines; LAYERED-TPOT and TEAPOT-SH variants

improve efficiency via hierarchical search (evaluating pipelines on increasingly large data subsets) and successive halving (Jamieson and Talwalkar, 2016).

EDCA (Evolutionary Data-Centric AutoML) (Simões and Correia, 2025) is an efficient AutoML solution that optimizes the entire ML pipeline. It performs data analysis, followed by the selection and optimization of data preparation steps and a model, using a genetic algorithm.

ML-Plan (& ML2-Plan) (Mohr et al., 2018; Wever et al., 2019) is a hierarchical planning-based (Erol et al., 1994) AutoML system for constructing ML pipelines; ML2-Plan adapts the method for multi-label data.

DeepLine (Heffetz et al., 2020) is an AutoML tool that relies on meta-learning, deep reinforcement learning (RL) (Kaelbling et al., 1996; Sutton, 1998) and *hierarchical actions filtering* to generate machine learning pipelines.

AlphaD3M (Lopez et al., 2023) is an AutoML library that automates data pipeline and model optimization by combining deep reinforcement learning and meta-learning. It is part of a broader project (D3M) aiming to automate data science.

Auto-sklearn (& 2.0) (Feurer et al., 2015, 2022) is a scikit-learn-based AutoML system using Bayesian optimization via SMAC (Lindauer et al., 2022) and meta-learning; v2.0 improves efficiency with early stopping via successive halving and with task-specific portfolios (pre-defined pipelines).

GAMA (General Automated Machine Learning Assistant) (Gijsbers and Vanschoren, 2021) is an AutoML tool using genetic algorithms to evolve and ensemble machine learning pipelines asynchronously.

Auto-PyTorch (Zimmer et al., 2021) is an AutoML toolkit for deep learning using PyTorch (Paszke et al., 2019), automating neural architecture and hyperparameter search for tabular, time series, and image data, by using Bayesian optimization, multi-fidelity (Forrester et al., 2007; Klein et al., 2017; Falkner et al., 2018), meta-learning, and ensemble construction.

FEDOT (Nikitin et al., 2022) is an AutoML framework employing meta-heuristic evolutionary methods for learning and optimization on graph-based pipelines. FEDOT can handle tabular and multimodal data, with text, image, and time series.

## 6.2 Analysis

We proceed to an analysis of data preparation in the selected AutoML systems, supported by a comparative table.

### 6.2.1 Setup

Table 2 presents a breakdown of automated data preparation elements that are implemented in our list of AutoML systems, organized according to transformation categories, functions, and purposes, as described in our taxonomy in Section 4. In the last column, titled *DP Optimization*, we differentiate between systems with data treatments that are fixed or arbitrarily decided, and ones that optimize a data preparation pipeline along with the model. We order the systems based on (1) whether they perform optimized data preparation, (2) the number of transformation categories they cover, and (3) their publication date.

We consider only explicit data preparation operations, as opposed to ones implicitly performed by certain models in the model selection pool of an AutoML system (e.g., some tree models inwardly doing their own feature selection). The information in the table reflects the state of each AutoML system to the best of our knowledge, according to the contents of their corresponding research papers, documentation, and code.

### 6.2.2 Findings

Table 2 demonstrates that, barring a few exceptions, automated data preparation in AutoML systems today is by and large focused on model performance optimization, in line with the traditional model-centric trend in ML and AutoML in general.

The most frequently present transformations are feature engineering ones (feature generation, feature selection, and dimensionality reduction) and feature-oriented preprocessing ones (scaling and encoding), emphasizing the prioritization of model performance, via features in particular. The only other consistently represented transformation categories are the cleaning of missing values, which can completely obstruct the function of many models, and sometimes structural standardization. These indicate added consideration for the most common obstacles to a functioning model.

There is a notable lack of transformations centered around data quality, excepting the cleaning of missing values. Likewise, more complex data integration (dataset merging, semantic harmonization) going beyond the minimum of loading and interpreting data types, is almost entirely absent. Preprocessing transformations dealing with data points as opposed to features are also underrepresented. This reveals a tendency to rely on the model's capabilities to overcome potential data issues such as noise, inconsistencies, or imbalance, while only treating data as far as making it viable for use.

From a methodology point of view, the table exhibits a mix of AutoML systems that include data preparation in their optimization process, and those that apply preset transformations or follow a rule-based logic. Research-oriented solutions appear to lean towards optimization more than their production-oriented counterparts, potentially leading to questions about the theoretical versus practical efficiency and effectiveness of the two approaches.

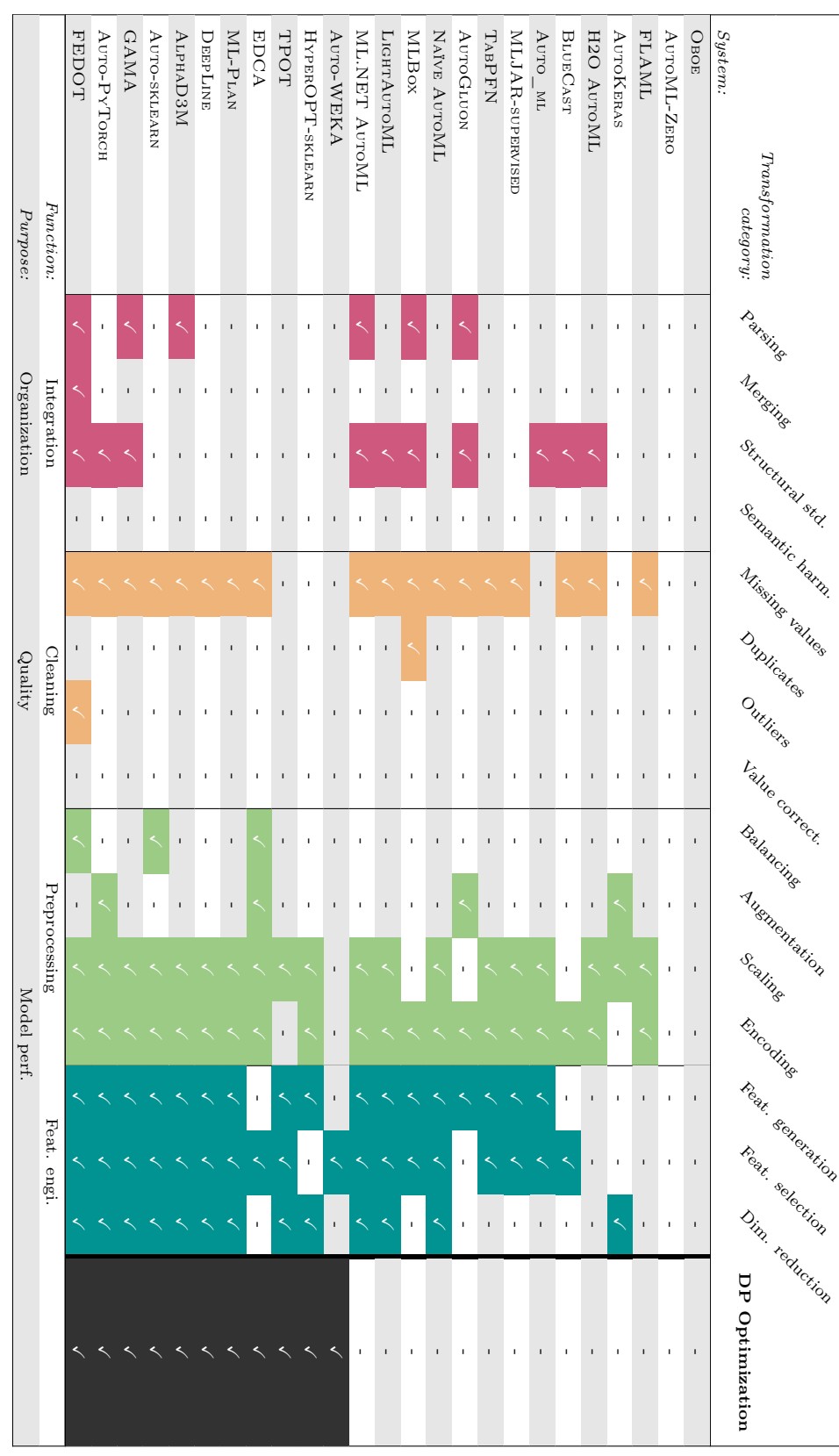

Table 2: Automated data preparation in AutoML systems, ordered by data preparation optimization, transformation categories, and publication date

## 7 Semi-Automated Data Preparation

Semi-automated data preparation encompasses approaches that partially automate the data pipeline or fully automate a particular part of it. While some aspects of data preparation can be automated, it has been suggested that others require human intervention (De Bie et al., 2022). Beyond expanding upon the set of transformation categories that can be automated, as seen in Section 6.2, we further seek to gain a better understanding of the importance of human contributions in the data preparation process, and whether the human component presents an obstacle to fully automating the pipeline.

In the following subsections, we provide an overview of various semi-automated methodologies, structured according to the different ways in which they integrate automated components, and the role of the human within their workflows. Though not all of these approaches are aimed specifically at machine learning applications, they can nonetheless deliver significant value in an ML context.

As a great many semi-automated tools and approaches exist in literature, it is quite possible that our collection may not be exhaustive. However, it provides a comprehensive overview, from well-known frameworks to newer methods, with a variety of semi-automation approaches which, put together, cover virtually all the data preparation categories presented in Section 4.

### 7.1 Full Automation of Parts of the Pipeline

The tools and frameworks below are specialized in automatically treating select parts of the data preparation pipeline.

COLNET (Chen et al., 2019) automatically annotates table columns with semantic types by predicting them based on column contents and a knowledge base, using convolutional neural networks (CNN) (LeCun et al., 2015) along with semantic embeddings (Camacho-Collados and Pilehvar, 2018).

ARCHETYPE (Feuer et al., 2024) is a framework for semantic column annotation using large language models (LLM) (Vaswani et al., 2017; Minaee et al., 2025). It can assign column types in a zero-shot manner, without requiring a pre-established set of semantic types, by relying on a pipeline of context sampling, prompt serialization, model querying, and label remapping.

UNIDM (a Unified framework for Data Manipulation) (Qian et al., 2024; Lin et al., 2025) leverages LLMs for various data preparation tasks. It breaks automated data manipulation down into three steps with prompts specifically designed for them: context retrieval to capture knowledge from data lakes, context data parsing to reshape the knowledge into natural language, and target prompt construction to design and effective LLM prompt for the data preparation task at hand.

SKRUB (skrub, 2018) is a Python library that aims to "directly connect database tables to machine learning estimators". It offers advanced automated tools that smartly handle

multiple complex data transformation categories, going beyond the basic transformations found in more typical data preparation solutions. The library is still under development, with additions of new submodules for a wider range of functionalities.

ExploreKit (Katz et al., 2016) performs automatic feature generation. It generates a large number of features by combining existing ones, then uses its ML-based feature selection process to predict the usefulness of the new features and to select the best ones.

AutoLearn (Kaul et al., 2017) is a regression-based feature learning algorithm for automated feature generation and selection. It analyzes pairwise feature relationships, then generates new features based on discovered insights, and finally makes a selection by finding the best predictors via regression.

TSFresh (Christ et al., 2018) is a Python library for automatic time series feature extraction and selection. It generates features using pre-defined time series characterization methods, then makes a selection via hypothesis testing, using $p$-value calculation with multiple testing.

FeatLLM (Han et al., 2024) is a LLM-based system that performs automated feature engineering. It makes use of LLM capabilities to generates prompts for feature generation augmented by domain knowledge, as well as for informed feature selection.

## 7.2 Human-in-the-Loop Approaches

Human-in-the-loop approaches refer to ones that include both automated processes and the involvement of a human user. The methods in this category can be subdivided into single-interaction systems, where the user only intervenes at the beginning of the process while the rest of it is fully automated, and systems that operate in an interactive loop of human and automated elements, constructing their pipeline in an iterative process.

### 7.2.1 Single Interaction

The methods below enable automated data preparation with the help of a single human intervention that provides insights into the data or examples of data treatment to learn from. As such, they can also be classified as human-guided automation methods.

FlashExtract (Le and Gulwani, 2014) automatically extracts raw data from text files, webpages, and spreadsheets. Using human-generated examples, it synthesizes extraction scripts in a domain-specific language.

The *Deep Feature Synthesis* (Kanter and Veeramachaneni, 2015) algorithm and the supporting FeatureTools tool are geared towards automated feature creation. They extract feature tables from relational database-type structures.

SampleClean (Krishnan et al., 2015) proposes a method consisting in manually cleaning only a sample of database data, then using it to automatically estimate aggregate query results on dirty data. It is accompanied by ActiveClean (Krishnan et al., 2016), which extends the concept to some ML models, by selecting data to prioritize cleaning for during training. BoostClean (Krishnan et al., 2017) takes this a step further, identifying both the

data to clean and a cleaning method for it, using statistical boosting (Schapire, 1990, 2003) and *Word2Vec* word embeddings (Mikolov et al., 2013). Lastly, AlphaClean (Krishnan and Wu, 2019) fully automates the design and parameter configuration of entire data cleaning pipelines.

*Data programming* (Ratner et al., 2016) presents a paradigm for creating and modeling labeled training datasets. For an unlabeled or partially labeled dataset, users encode weak supervision by way of *labeling functions*, which each provide a label for a subset of the data. The collection of labeling functions forms a generative model, allowing the dataset to be denoised through making decisions on overlapping labels, by learning the accuracies and correlation structure of the labeling functions.

HoloClean (Rekatsinas et al., 2017) is a data cleaning framework for value correction using probabilistic inference (Neal, 1993; Box, 2011). It relies on human input in the form of constraints and metadata, and can also accept feedback on its data repairs.

CPClean (Karlaš et al., 2020) performs data cleaning with the aid of a human who provides ground truth information. Its algorithm is rooted in the concept of *Certain Predictions*, aiming to correct the data in such a way that any predictor would give the same prediction results.

Raha (Mahdavi et al., 2019) and Baran (Mahdavi and Abedjan, 2020) are tools for value error detection and correction, respectively. The detection system is configuration-free and relies on the generation of feature vectors to cover various types of data errors. The correction system uses transfer learning for multiple error correction, with the aid of context-aware data representation. The systems are highly automated, requiring only a few initial user-provided examples.

HAIPipe (Chen et al., 2023) is a system combining human-generated pipelines (HI-pipelines), designed with domain knowledge and experience, and machine-generated ones (AI-pipelines), which are search-based and optimized, into a best of both worlds solution. It uses an *enumeration-sampling strategy* to find the best combined pipeline: enumerating possible pipelines and using active learning (Cohn et al., 1994) to select the best-performing one.

### 7.2.2 Interactive Loop: Human-Guided Automation

An interactive loop implies back-and-forth exchanges between human and automation in the pipeline optimization process.

Interactive approaches also present multiple examples of human-guided automation, where the design of the data pipeline is mostly automated, but with the guidance of a human user. The user can contribute in different ways depending on the system, such as analyzing the data, highlighting areas that require attention, or providing feedback on results.

*Feedback-driven improvement* of data preparation pipelines (Konstantinou and Paton, 2020) is an approach where the user guides an otherwise automatic data preparation process, by

providing feedback regarding resulting data after transformations are applied. The provided feedback represents correctness criteria.

Rain (Wu et al., 2020) is a complaint-driven training data debugging system. It uses human feedback on dataset queries to resolve issues and improve dataset contents through heuristic approaches based on influence functions (Koh and Liang, 2017).

*Cleaning for ML* rather than before ML (Neutatz et al., 2021) is an interactive architecture proposal for cleaning data during the training process using model feedback, instead of cleaning independently before training. While the cleaning workflow is automatically generated and evaluated over time, the user is integrated into the loop to oversee the process and provide signals aiding decision-making, such as annotating errors or certain properties.

Alpine Meadow (Kraska, 2018; Shang et al., 2019) is an interactive AutoML tool providing a user-guided way of curating ML pipelines in an effort to simulate a data scientist's process. The system combines query optimization ideas and pipeline optimization methods such as Multi-Armed Bandits (Robbins, 1952; Slivkins, 2019), Bayesian optimization (Snoek et al., 2012; Garnett, 2022), and meta-learning (Schmidhuber, 1987; Hospedales et al., 2021), with human input in the form of feedback at every step.

The Automatic Statistician (Steinruecken et al., 2019) presents a vision for automated data science frameworks with minimal human intervention. Beginning at raw datasets, the concept covers data preparation, modeling, evaluation and insights. It also includes considerations for explainability and resource budget. Interactions with users in natural language are envisioned through automatically generated reports. The user guides the system by indicating certain preferences such as model and evaluation methods, areas of interest in the data, or resource constraints. Every other aspect of the process is automated.

Learn2Clean+HIL (Learn2Clean with Human-In-The-Loop) (Berti-Equille, 2020) is an extension of the automated reinforcement-learning-based (Kaelbling et al., 1996; Sutton, 1998) data preparation system Learn2Clean (Berti-Equille, 2019). In this variant, a human user is integrated into the process in order to actively give regular feedback to the system through a manual mechanism for RL rewards.

### 7.2.3 Interactive Loop: Human Assisted by Automation

Another kind of approach interactively integrating human and automation is one where the roles are reversed: a user constructs a data preparation pipeline with the assistance of an automated system. In this format, the automation usually includes a preliminary data analysis or an optimization process, in order to provide recommendations to the user that help build or improve the data pipeline.

Data Diff (Sutton et al., 2018) performs automated data wrangling on data that is periodically resampled, by detecting certain types of differences (inconsistencies/corruptions) and *patching* (fixing) them using transformations in a domain-specific language. If an automatic patch is not feasible, it informs the user.

Auto-Prep (Bilal et al., 2022) is a partially automated data preparation solution that detects the data problem (ML task), generates visualizations, and makes data cleaning and preparation recommendations. The user then performs actions based on those insights.

Predictive Interaction (Heer et al., 2015) is a framework design that provides data visualizations, lets the user choose features of interest from them, and then uses predictive methods to suggest data transformations accordingly. This process is repeated in a loop where the system provides guidance while the user makes decisions.

ChatPipe (Chen et al., 2024) allows a user to design a data preparation pipeline through interactions with the *ChatGPT* (Radford et al., 2018; Brown et al., 2020; OpenAI et al., 2024) LLM. The system creates ChatGPT prompts based on the problem and wishes expressed by the user, who in turn receives recommendations for subsequent data preparation transformations, as well as automatically generated code.

Another LLM-based approach is that of interactively improving data preparation code by automatically generating *shadow pipelines* (Grafberger et al., 2024). It consists in generating variants of an original pipeline that detect issues, try improvement modifications, and suggest and explain them to the user, using LLMs. The user can then choose to integrate these changes.

## 7.3 Analysis

Similarly to the previous section, we proceed to analyze semi-automated data preparation in different systems.

### 7.3.1 Setup

Table 3 reflects the data preparation coverage of the semi-automated systems above, relative to the transformation categories, functions, and purposes described in the taxonomy in Section 4. Three additional columns are included to illustrate the variety in approaches, outlining which tasks are undertaken by humans and which by automation, and how the two interact. The systems are ordered by (1) semi-automation approach, (2) transformation categories, and (3) publication date.

The *Guidance* column indicates whether the human or the automation (or both/neither) provides guidance to their counterpart within the data preparation process. This assistance can come in different forms, such as offering examples, data insights, or transformation recommendations.

The *Action-taking* column indicates whether the human or the automation (or either) has the final say in terms of making decisions and performing transformations on the data.

The *Interactive loop* column indicates whether the system involves an iterative back-and-forth process between human and automation while constructing a data preparation pipeline—as opposed to a fully automated approach, or a one-shot interactive approach where one actor (human or automation) participates at the beginning of the process, and the other actor then takes over without further interactions between the two.

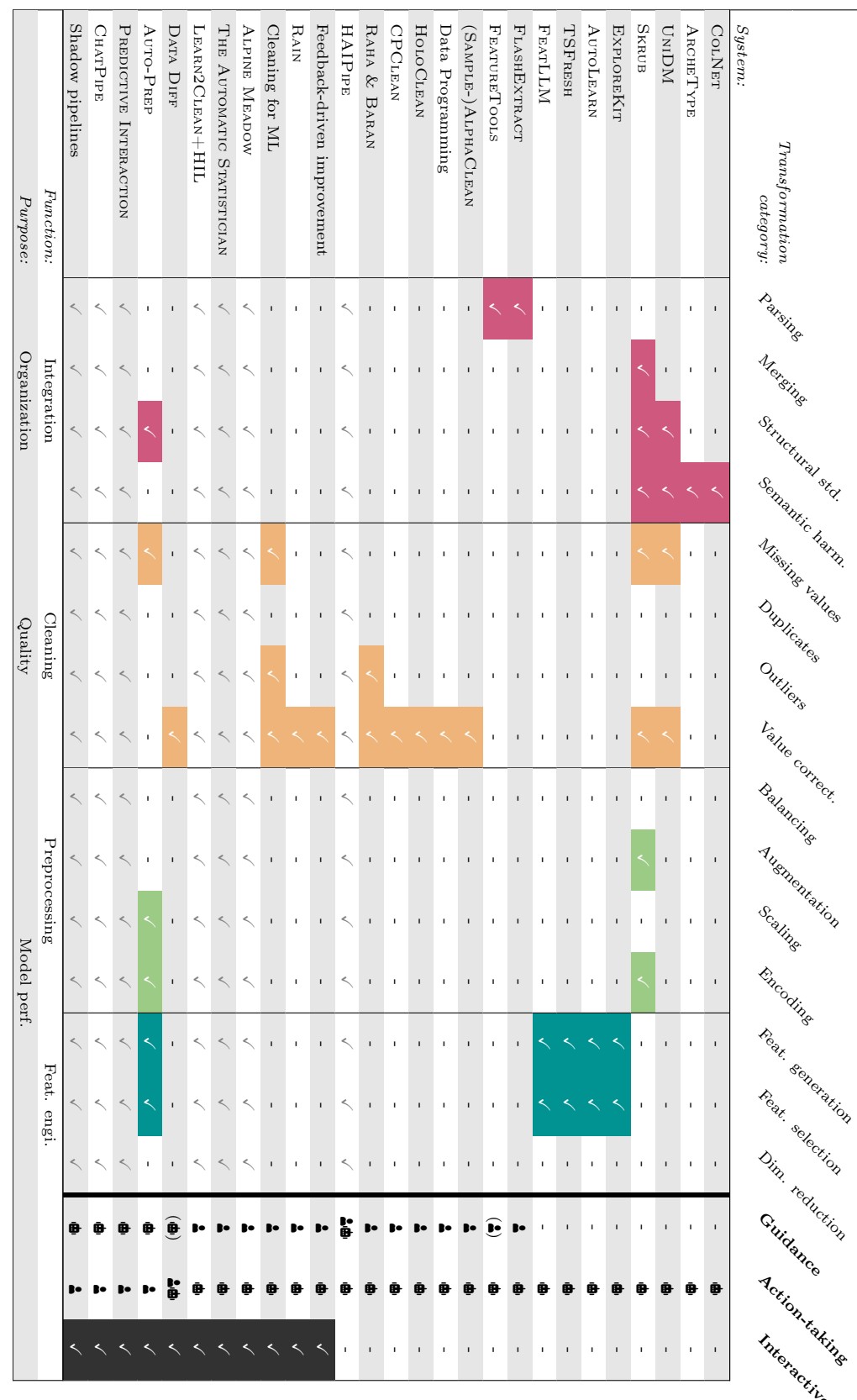

Table 3: Semi-automated data preparation approaches, ordered by semi-automation approach, transformation categories, and publication date (✓: *not implemented but the method allows it*)

### 7.3.2 FINDINGS

Regarding explicit pipeline coverage in terms of data transformation categories, we note that, with a few exceptions, the systems that actually implement transformations tend to concentrate on a single transformation function. On the one hand, we have data management tools centered around data integration or cleaning (mostly value correction), and on the other hand, tools focusing on feature engineering (feature generation and selection) from more ML-oriented works.

Looking back at the transformation categories covered by AutoML solutions (c.f. Section 6.2), where we saw that data preparation in AutoML was majorly centered around feature transformations, it is interesting to find that many semi-automated systems that do not include a human component likewise focus on feature manipulation. Conversely, semi-automated approaches show more inclination towards data integration and cleaning than their AutoML counterparts, especially in the areas of parsing, semantic harmonization, and value correction. These categories are plausibly where human insight is most needed, and may thus be the hardest to automate.

While fully automated approaches and some partly automated ones offer concrete implementations of both their methods and data transformations, others are more conceptual in nature—particularly when it comes to data transformations. Table 3 reflects this, as several frameworks propose methods with the potential to integrate the entire data preparation pipeline, without necessarily addressing the specifics of individual transformations. Instead, these systems primarily focus on pipeline design methodology.

### 7.3.3 FURTHER AUTOMATION PERSPECTIVES

The semi-automated solutions examined in this section present a variety of ways to combine human and automated components. Some necessitate regular human participation, yet others require it only once. Some rely on human data insights and automate pipeline construction, while in others these responsibilities are reversed. The main value of human input when it comes to data preparation is generally considered to be their domain knowledge and technical expertise. The design choices in semi-automated systems indicate that this input still holds value. Yet, the differing degrees of human intervention and possibility or role reversal with automation would suggest that any aspect of the data preparation process could selectively be automated—sustaining the prospect of automating all of them in a unified process. The question remains whether this is feasible without compromising results, i.e., whether automation alone can adequately compensate for human contributions.

## 8 Automated Data Preparation

Having explored the extent of data preparation in automated machine learning and examined semi-automated data preparation approaches, we now turn to fully automated data preparation. For this category of approaches, we take into consideration systems that are specialized in automating data pipeline optimization, as well as AutoML systems whose optimization process covers the data pipeline along with the model (c.f. Section 6.2).

Underlying methodologies for the optimization of data pipelines range from traditional optimization techniques, such as heuristic search and Bayesian optimization, to emerging ML paradigms such as foundation models. Some also integrate techniques for context-awareness (Chai et al., 2023a) through the use of language models (word embeddings, LLMs), or learning from experience (Giovanelli et al., 2022) via transfer or meta-learning, in order to make the process more effective and efficient. We review these different approaches with the aim of extracting a structured overview of relevant methods along with their advantages, limitations, and particularities. We then present a comparative summary of our findings.

## 8.1 Methodologies and Examples

Below are descriptions of optimization methodologies used for automated data preparation, along with examples of systems that employ them.

### 8.1.1 Sequential Model-Based Optimization

SMBO (Sequential Model-Based Optimization) (Jones et al., 1998; Bartz-Beielstein et al., 2005; Hutter et al., 2011) is one type of method that be used to automatically optimize data preparation pipelines. An SMBO method navigates its search space with the help of a *surrogate model* that approximates the objective function, i.e., in our context, a model evaluation metric. The surrogate model is built by fitting data pipelines to models, training them, and evaluating them. It iteratively selects new points to evaluate by optimizing an *acquisition function*, which balances exploration and exploitation. The surrogate model is updated at every step based on the newly evaluated point. The most commonly used SMBO method is Bayesian optimization (Snoek et al., 2012; Garnett, 2022), where the surrogate model is probabilistic, such as a Gaussian process (Rasmussen, 2004).

SMBO methods allow for efficient optimization while focusing on the most informative or promising areas of the search space. Depending on the underlying surrogate model, they may become inefficient at greater scale, though there have been advances to mitigate that effect (Eriksson et al., 2019). They are explicitly designed to support budget restrictions in the form of limited evaluations. Through the choice of interpretable surrogate models, they may in some cases provide explainability.

SMBO techniques have proven capable of automatically optimizing data pipelines with different surrogate functions—SVMs, neural networks, random forests, decision trees (Hastie et al., 2001)—in a fixed-architecture *DPSO* (Data Pipeline Selection and Optimization) (Quemy, 2019) setup.

There are also multiple examples of Bayesian optimization approaches among AutoML solutions that include data pipeline optimization, such as Auto-WEKA (Thornton et al., 2013; Kotthoff et al., 2017) and HyperOPT-sklearn (Komer et al., 2014). Others, namely Auto-PyTorch (Zimmer et al., 2021) and Auto-sklearn (Feurer et al., 2015, 2022), rely on a Bayesian optimization base algorithm augmented by combinations with other approaches.

### 8.1.2 Evolutionary Algorithms

Evolutionary algorithms (Bartz-Beielstein et al., 2014) are optimization methods inspired by biological evolution and natural selection processes. They can iteratively optimize data preparation pipelines by evolving a population of *candidate solutions* in the form of data pipelines, over successive *generations*. At each iteration, the pipelines are evaluated according to a *fitness function*, corresponding to the downstream model's evaluation metric. The best-performing candidates are retained. Those pipelines evolve into a next generation through mechanisms such as *mutation* via the addition, removal, swapping, or hyperparameter reconfiguration of data transformations, or *crossover* combinations with other pipelines.

Evolutionary approaches promote exploration, and are well-suited for complex black-box optimization problems. They can implement budget restrictions by limiting the population size and the number of generations, and offer some explainability by tracing back the evolution process. Depending on the optimization landscape and algorithm settings, their main limitation is that they may be slow to converge.

Evolutionary algorithms are the optimization method of choice for several AutoML systems with data pipeline optimization, namely FEDOT (Nikitin et al., 2022), GAMA (Gijsbers and Vanschoren, 2021), TPOT (Olson and Moore, 2016), and EDCA (Simões and Correia, 2025).

### 8.1.3 Hierarchical Planning

Hierarchical planning (Erol et al., 1994) draws on AI planning concepts to break down complex tasks into simpler subtasks. This can be done recursively, resulting in multiple levels of subtasks. Decisions are made at each level using search, typically best-first search (Pearl, 1984; Dechter and Pearl, 1985) or anytime search (Hansen and Zhou, 2007), and evaluation. The principle can be applied to build ML or data pipelines in a top-down manner, by treating the construction process as a hierarchical task decomposition problem (Erol et al., 1994). The highest level task would be the goal itself: pipeline optimization. It can then be decomposed into subtasks at multiple levels. For instance, one subtask could be transformation category selection, then at the next level transformation operation selection within a category, and then hyperparameter configuration.

Hierarchical planning provides a structured search approach, which offers transparency and interpretability. It evaluates partial pipelines and prunes weak performers, making it quite efficient. In addition, it can provide a solution at any time (but also further refine it), allowing for a form of budget control. Its drawbacks are its dependency on the quality of underlying heuristics, overhead that can result from recursive decomposition and search, and its sequential nature, which limits parallelism.

The ML-PLAN (Mohr et al., 2018; Wever et al., 2019) AutoML system uses hierarchical planning to build ML pipelines. Using a hierarchical task network (HTN) process, it decomposes the task into feature preprocessing and modeling decisions at multiple levels.

### 8.1.4 GRADIENT DESCENT

Gradient descent (Hastie et al., 2001; Boyd, 2006) algorithms, very common in ML, optimize model parameters by iteratively calculating the gradient of the loss function and adjusting parameter values so as to minimize the loss. Though they are typically used for the downstream ML model's parameters rather than data preparation pipeline optimization, gradient descent algorithms can also be employed for the latter, as long as the pipeline construction process is modeled to be differentiable. The main benefit of using gradient descent for this problem is that it can optimize the data pipeline along with the model as a bi-level optimization problem, thus training the model only once—unlike most other methods that require repeated training with different pipelines.

Gradient descent is a very efficient optimization approach, but only works with differentiable downstream ML models. It also bears the risk of the gradient getting stuck in local minima, thus a proper tuning of hyperparameters, in particular the learning rate, is important. Gradient descent additionally allows for limited budget considerations through techniques such as early stopping. Below are two instances of gradient-based systems.

DIFFML (Hilprecht et al., 2023) is an approach that consists in making whole ML pipelines, including data preparation pipelines, differentiable. It achieves this by expressing different pipeline possibilities as a so-called *mixture* of pipeline alternatives, whose weights can be optimized during the training process. By making the complete machine learning pipeline differentiable, this approach allows for combined data pipeline and model optimization with gradient methods.

DIFFPREP (Li et al., 2023) for tabular data is a method for data preparation pipeline searching along with model optimization, for differentiable ML models. It treats the problem as a bi-level optimization problem, minimizing training loss for the model training, and validation loss for the data pipeline design process. By relaxing the discrete non-differentiable pipeline search space into a continuous differentiable one (Liu et al., 2019), this approach allows for pipeline optimization via gradient descent, jointly with model optimization.

### 8.1.5 REINFORCEMENT LEARNING

Reinforcement learning (Kaelbling et al., 1996; Sutton, 1998) systems learn by performing *actions* in a modeled stateful *environment*. They receive environment feedback for those actions in the form of *rewards* following changes of *state*, and adjust their future behavior accordingly—thus iteratively shaping their decision *policy*. In the case of data preparation pipeline construction, actions correspond to data transformation decisions, while rewards represent downstream model performance. The developed decision policy is what guides pipeline construction, and the result is an optimized data pipeline.

RL approaches are typically good at modeling sequential tasks and learning long-term consequences of actions. However, they often require a large number of training loops, and may not converge in a stable manner. Below are a few examples of RL approaches.

Learn2Clean (Berti-Equille, 2019) relies on the Q-learning (Watkins, 1989; Watkins and Dayan, 1992) reinforcement learning technique to perform data preparation on Web data, which usually implies large amounts of dirty data. The system takes a dataset, ML model, and an evaluation metric as input. The choice of Q-learning allows for a relatively lightweight RL system despite the large search space due to its *model-free* nature, as well as a flexible one thanks to frequent reward updates for performed actions. Q-learning is typically effective on small search spaces, but may not always scale well.

HAIPipe (Chen et al., 2023) is a semi-automated approach to data preparation, that includes automated data pipelines called *AI-Pipes* as one of its components—the other being a human-generated pipeline to combine with it. For our purposes, we focus on how this approach generates AI-Pipes. It proposes a method for the iterative design of pipelines based on DQN (Deep Q-Network) (Mnih et al., 2015) reinforcement learning. The DQN technique offers better scalability to large state spaces than tabular Q-learning, but is more computationally expensive.

CTXPipe (Gao et al., 2024) is a system for context-aware automated data pipeline design. It is able to integrate additional context into the process by using pre-trained semantic embedding models (Camacho-Collados and Pilehvar, 2018). This approach provides the system with domain expertise by allowing it to augment its training data through the extraction of semantic insights from it. With this added information, the system constructs data preparation pipelines using DQN reinforcement learning as its optimization method.

We can also find AutoML systems that optimize the data pipeline with reinforcement learning, such as DeepLine (Heffetz et al., 2020), which uses other techniques in addition RL to further enhance its optimization process.

### 8.1.6 Meta-Learning

Meta-learning (Schmidhuber, 1987; Hospedales et al., 2021) is a process of *learning to learn* through prior experience. In the case of data preparation, by training many different data pipelines on many different datasets, meta-learning methods can capture knowledge about the pipeline optimization process by extracting common patterns. This provides them with strong generalization capabilities and fast convergence. Instead of learning how to optimize a pipeline for each dataset from scratch, meta-learning systems can leverage their gathered knowledge to efficiently construct data pipelines for previously unseen datasets. Nonetheless, the overhead in training a meta-learning model can be quite heavy.

One solution for automated data preparation via meta-learning (Bilalli et al., 2016) employs meta-learning concepts to evaluate positive or negative effects of different data preparation transformation applications within a ML pipeline. It then provides a *transformation ranking*, i.e., a set of recommended transformations to apply to the data.

Another, MetaPrep (Zagatti et al., 2021) is an automated data preparation system based on meta-learning. Trained on a collection of example datasets and data pipelines, the system learns by extracting meta-knowledge in the form of data characteristics, model evaluations after applying data pipelines, and dataset similarity indicators. It can subsequently apply

that knowledge to create quality data preparation pipelines for new datasets. MetaPrep returns the five best pipelines among those generated.

### 8.1.7 Foundation Models

Foundation models (Bommasani et al., 2022), such as large language models (LLM) (Vaswani et al., 2017; Minaee et al., 2025), are large pre-trained models that are trained on massive amounts of data. They offer general-purpose utility for a wide range of downstream tasks. Using vast corpora of language data from a great diversity of domains, LLMs learn broad language patterns that allow them to restore their knowledge by generating natural language, as well as programming code. For the purpose of building data preparation pipelines, LLMs can recommend data transformations and provide corresponding code. They can also optimize pipelines through repeated interaction to further improve their output.

LLMs offer unique upside in their ability to access domain knowledge outside of the user-provided data. However, LLMs also come with some known downsides: they are initially *very* expensive to train, their inference process is costly, and their generated output is not always reliable—they sometimes produce inconsistent responses, hallucinations, or inefficient code (Hadi et al., 2023; Laskar et al., 2024; Michelutti et al., 2024; Kostikova et al., 2025). A typical observation is that LLMs can generate good solutions with far fewer trials than classical optimization methods, but do not reach the same final performance.

CAAFE (Context-Aware Automated Feature Engineering) (Hollmann et al., 2023) presents an LLM-based approach to automated data preparation that includes data context awareness. The system takes a dataset accompanied by a natural language problem specification by a user as input. It then iteratively generates a data preparation pipeline along with corresponding code, applies it to the data and trains a model, then re-generates data preparation code based on the model's performance results, and continues repeating the process.

### 8.2 Combined Methodologies

Combining multiple optimization methods is a convenient way to take advantage of their complementarity in order to enhance results. It is also possible to break down the optimization process and select a method most appropriate for each part.

Saga (Siddiqi et al., 2023) is a data preparation framework that builds data pipelines in two phases. It starts by finding the most promising pipelines using an evolutionary algorithm, which optimizes pipelines by adding or removing data transformations from them, reordering them, or mixing parts of two pipelines. The data transformations in this step are all configured with default hyperparameters. In the second phase, transformation hyperparameters are tuned using successive halving with Hyperband (Jamieson and Talwalkar, 2016; Li et al., 2018). Finally, the system ranks the pipelines and returns the top-$k$ ones. Saga is designed to support parallelization. It also optionally includes user input in the form of custom constraints or data transformations.

AutoML systems also present a variety of combinations of optimization algorithms.

Auto-PyTorch enhances its Bayesian optimization approach with multi-fidelity optimization (Forrester et al., 2007; Klein et al., 2017; Falkner et al., 2018) to strategically allocate its evaluation budget, and with meta-learning in order to warm-start the Bayesian optimizer.

Similarly, Auto-sklearn, based on Bayesian optimization, uses meta-learning to initialize its Bayesian optimizer in order to start evaluations from quality points, and employs successive halving for better budget allocation.

Two members of the TPOT family extend the base approach with different additions: Layered-TPOT (Gijsbers et al., 2017) introduces a hierarchical search method that creates a layered structure to reduce the number of evaluations by favoring promising pipelines. TPOT-SH (Parmentier et al., 2019) uses successive halving to balance its budget, allocating less time to weak pipelines, and more to promising ones.

DeepLine constructs machine learning pipelines using reinforcement learning, augmented by meta-learning to guide actions, and hierarchical action filtering to structure and prune the search space, leading to faster convergence.

These combined approaches demonstrate the benefits of using complementary optimization techniques to enhance or overcome certain shortcomings of the base optimization algorithm. The most frequent choices are the use of meta-learning to warm-start the search process, thus speeding it up and increasing the likelihood of a quality solution; hierarchical methods to structure the search process, making it more efficient; multi-fidelity optimization and in particular successive halving, to improve search budget allocation based on the pertinence of the explored solution. Naturally, the addition of these techniques also comes with a cost, however, it is worth considering whether the extra cost is offset by their contributions.

### 8.3 Comparative Overview

Table 4 presents a comparative overview of pipeline optimization methods for automated data preparation. It summarizes their main advantages and limitations, and rates them according to relevant criteria when selecting an approach: (1) *effectiveness* in terms of final solution quality; (2) *efficiency*, both sampling and computational; (3) having *low pre-requirements* such as additional resources and overhead; (4) *context-awareness*; (5) time, resource, or evaluation *budget control*; and (6) *explainability*. It also categorizes the individual approaches discussed in the subsections above based on the broader optimization methodology they belong to. AutoML solutions (that also optimize the model, as opposed to just the data pipeline) are marked with an asterisk (*). Solutions that combine multiple optimization methods are marked with (+).

It is worth noting that the mentioned advantages and limitations are of a rather general nature, and that individual approaches may implement additional techniques to overcome certain shortcomings. Similarly, the ratings across several criteria represent estimates rooted in broadly known qualities of each methodology, but practical results may differ depending on factors such as data characteristics or method implementations.

| Opt. method | Advantages | Limitations | Effectiveness | Efficiency | Low pre-req. | Context-aware. | Budget control | Explainability | Example systems |
|---|---|---|---|---|---|---|---|---|---|
| SMBO (e.g., Bayesian optimization) | • Good at black-box optimization
• Sample-efficient
• Surrogates can capture uncertainty | • Poorer scaling
• Relatively large overhead | ✓ | ✓ | ~ | - | ✓ | ~ | DPSO
AUTO-WEKA(*)
HYPEROPT-SKLEARN(*)
AUTO-PYTORCH(*)(+)
AUTO-SKLEARN(*)(+) |
| Evolutionary algorithms | • Good at black-box optimization
• Strong global exploration | • Slow convergence | ✓ | ~ | ✓ | - | ~ | ~ | FEDOT(*)
GAMA(*)
TPOT(*)
EDCA(*)
SAGA(+) |
| [Opt. +] Hierarchical planning/search | • Structured and interpretable search
• Efficient pruning
• Anytime optimization | • Heuristic dependency
• Planning overhead | ~ | ✓✓ | ~ | - | ~ | ✓ | ML-PLAN(*)
LAYERED-TPOT(*)(+) |
| Gradient descent | • Fast
• Efficient | • Directly usable on differentiable problems only
• Can get stuck in local optima | ✓ | ✓✓ | ~ | - | - | - | DIFFML
DIFFPREP |
| Reinforcement learning | • Good at sequential tasks
• Learns long-term effects of actions | • Sample-inefficient
• Unstable convergence | ✓ | ~ | ~ | - | - | - | LEARN2CLEAN
HAIPIPE
CTXPIPE(+)
DEEPLINE(*)(+) |
| [Opt. +] Meta-learning | • Generalization across datasets and pipelines
• Few-shot adaptation
• Warm-starting optimizers | • Expensive training
• Requires many prior datasets and pipelines | ✓ | ✓✓ | ✗ | - | - | - | *Transformation ranking*
METAPREP
AUTO-PYTORCH(*)(+)
AUTO-SKLEARN(*)(+) |
| Foundation models (e.g., LLMs) | • Provides domain knowledge
• Query-efficient | • Unreliable
• Requires pre-trained foundation models
• Sub-optimal solutions | ~ | ✗ | ✗✗ | ✓✓ | - | - | CAAFE |
| Opt. + Multi-fidelity (e.g., Successive halving) | • Budget allocation balancing | • Requires reliable fidelity estimates | ✓ | ✓✓ | ~ | - | ✓ | - | SAGA(+)
AUTO-PYTORCH(*)(+)
AUTO-SKLEARN(*)(+)
TPOT-SH(*)(+) |
| Opt. + Semantic embeddings | • Provides domain knowledge | • Requires pre-trained embedding models | ✓ | ~ | ✗ | ✓ | - | - | CTXPIPE(+) |

Table 4: Automated data preparation approaches
((*): *AutoML system*; (+): *combined methods*)

Despite the lesser prominence of data preparation in AutoML when compared to modeling, the contributions reviewed in this section nonetheless show significant progress in automating data pipeline optimization. They offer a varied selection of approaches depending on needs and available resources, as can be seen through comparison along some meaningful criteria in Table 4. These approaches propose different compromises in terms of efficiency, pre-requirements, as well budget or explainability concerns. The context-aware nature of certain methods is of particular interest, since it adds a new dimension to the automated process by providing a degree of domain expertise. Usually reserved to humans, this aspect represents one of the main challenges in effectively automating data preparation, next to achieving an efficient pipeline optimization process.

## 9 Conclusions and Outlook

*Summary and Discussion.* Traditionally, performance optimization efforts in machine learning, as well as AutoML, have been concentrated on the modeling aspect of the workflow. The recent rise of the field of data-centric AI has raised awareness about the merits of optimizing data, in order to improve not only model metrics, but also quality in terms of faithfully representing real-world phenomena. This entails a growing interest in optimizing data preparation ahead of, or in combination with, model optimization.

Data preparation is essential to making raw data usable and effective for machine learning tasks. Our structured overview of data transformations according to their purposes, functions, and category types illustrates the plethora of possible configurations in data pipeline optimization. In practice, selecting transformations, ordering them, and optimizing their hyperparameters is a time- and labor-intensive procedure. Handling it with automation allows to streamline the decision process, and substantially alleviates the required effort.

Examining data preparation in existing AutoML solutions has shown that they typically rely on their input data being ML-ready beforehand. We found that those systems, relatively light on data treatment, are primarily focused on basic usability and feature engineering, overlooking other important aspects such as data quality. Furthermore, only a few AutoML solutions include a pipeline optimization architecture that could support the complete data preparation process. Even so, none of them fully implement such a workflow.

Among semi-automated approaches, the human-guided-automation and automation-assisted-human method alternatives for pipeline construction highlight the persistent value of human analytics and domain expertise. The different roles of human and automation in these approaches allow for a view into the difficulties of automating certain data preparation elements, notably those that require domain knowledge, such as parsing and semantic harmonization. At the same time, they present an opening to further automate the process.

In a review of fully automated data preparation approaches and their varied underlying optimization methods, we comparatively assessed their advantages and limitations. They namely consist in tradeoffs between effective, efficient optimization and resource requirements, in addition to considerations of budget restrictions and explainability. Moreover,

the inclusion of natural language techniques to provide context-awareness to certain data pipeline optimization methods has shown promise in terms of replicating the benefits of domain expertise in an automated fashion.

*Challenges, Opportunities, Future Directions.* The main challenge in automating data preparation for ML remains the design of an optimization algorithm for the vast search space of data pipeline configurations that is both efficient and effective. A variety of optimization methods have been proposed to tackle this problem, yet, despite their merits, these approaches leave room for further refinement. A recurring difficulty is that of integrating useful domain knowledge into the decision process without significant overhead. Moreover, pipeline optimization under different constraints calls for additional considerations and adapted solutions. Seeking data- and problem-specific insights may help with streamlining the optimization process.

The design of optimal data pipelines is tightly coupled with another problem—that of reliably evaluating and benchmarking them. The common use of downstream task evaluation as a proxy has proved to be an effective method, however, it can be quite inefficient, and is not perfectly representative. A more direct data pipeline assessment method for a given task could potentially lead to a more accurate, reliable, and efficient optimization process. Additionally, establishing a standard set of benchmarks would hopefully allow for a relatively objective empirical comparison between pipeline optimization methods.

Since its origins, AutoML research has largely been centered around tabular data. More recently, however, it has also been branching out to an increasing number of data modalities, which naturally extends to new automated data preparation needs. While pipeline optimization methods may be modality-independent, this raises the question of whether they could be tailored to certain particularities of different data modalities, creating avenues for the development of more specialized approaches.

Though select areas of data treatment have successfully been automated by the data management or machine learning communities, mainly in the areas of data organization and quality for the former, and model performance optimization for the latter, a separation between the two is quite evident. Beyond the search for novel techniques, connecting the contributions of both communities in a single pipeline architecture would allow to bridge that gap and implement a more holistic process that is automated end-to-end, ranging from raw data to ML model predictions.

The emerging data-centricity paradigm in ML has brought to light the unrealized potential in optimizing data to expand the limits of machine learning processes. There has been significant progress in automating data preparation pipelines to date, yet substantial challenges still remain. Continued research in these directions, and the eventual complete automation of the ML workflow from raw data to inference, would constitute a profound advancement towards automating data science in an increasingly data-driven world.

## Broader Impact Statement

As a survey and synthesis of existing literature, this work likely does not entail profound societal consequences in itself. However, research on the topic of the survey—automating data preparation for machine learning—could potentially have tangible effects on data science and machine learning common practices, resulting outcomes, and, to a lesser degree, the environment.

Similarly to AutoML, automating data preparation lowers technical barriers, providing wider accessibility to data preparation and by extension machine learning, in a step towards democratizing AI. At the same time, it lessens the workload of skilled practitioners, allowing them to focus on other critical parts of their workflow instead, while reliably maintaining or improving performance. Optimized automation reduces the risk of human error, as well as the amounts of time and resources required for data preparation.

Conversely, automated data preparation solutions have not yet fully succeeded in replicating the value of human input to the process, notably in terms of domain knowledge. In addition, automation without human supervision can accentuate biases in the data, and lack of transparency in the decision process can be a limiting factor for high-stakes applications, such as in healthcare. Our work briefly addresses fairness and explainability concerns, but they are not its main focus. Hence, we would caution against over-reliance on automation in sensitive contexts.

## Acknowledgments and Disclosure of Funding

This research was jointly funded by the French National Research Agency (ANR-23-CE23-0035) and the German Research Foundation (DFG; LI 2801/7-1), through project OPT4DAC.

The authors would like to thank Professor Juliana Freire for her insightful feedback on this work. We are also grateful to the DMLR reviewers for their guidance in improving the quality of the paper.

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

## Appendix A. ML Terminology

| | |
|---|---|
| *Machine learning* | *Machine learning (ML)* is a subfield of artificial intelligence and of data science, focused on developing methods that can learn patterns from data in order to make predictions. |
| *(Learning) Algorithm* | A learning *algorithm* is the underlying procedure or method used to learn from data. Some common types of learning algorithms are neural networks, random forest, and *k*-nearest neighbors. In some papers, it is also called *inducer*. |
| *Hyperparameter* | A *hyperparameter* is an algorithm configuration variable that controls one of several aspects of the learning process (e.g., the learning rate—the magnitude of learning steps, which controls how fast new information overrides old information). |
| *(ML) Model* | A machine learning *model* is a system trained using a learning algorithm. It retains learned patterns and can use them to make predictions. |
| *Inference* | *Inference* is the process through which a trained model makes predictions on new data. |
| *Task* | A machine learning *task* is a specific problem to be solved by an ML model. Tasks can belong to different learning paradigms: supervised learning, where the model's training is guided by some known outcomes (e.g., regression—predicting values, or classification—affecting input data to known categories); unsupervised learning, where the model identifies patterns without external guidance (e.g., clustering—identifying groups of similar elements within the data); and others. |
| *Dataset* | A *dataset* is a structured collection of data, typically organized (for tabular data) into rows and columns. |
| *Data point* | A *data point* is a single observation, or instance of information. Data points usually correspond to the rows of a tabular dataset. |
| *Feature* | A *feature* is a property or variable describing data points. Features usually correspond to the columns of a tabular dataset. Features can belong to one of several types, such as numeric (e.g., age) or categorical (e.g., gender). |
| *Target variable / label* | A *target variable*, or *label*, is the outcome variable that a machine learning model is trained to predict. |
| *Training set* | A *training* dataset is a subset of data used to train a machine learning model. It is the dataset that the model learns on. |
| *Test set* | A *test* dataset is a subset of data used to evaluate the performance of a trained model after training is complete. It is kept separate from the training dataset, and is not used for learning. |
| *Validation set* | A *validation* dataset is an optional subset of data that can be used to evaluate the performance of a model during training, in order to adjust hyperparameters. It is kept separate from the training and test datasets, and is not used for learning. |
| *Metric* | A *metric* is a mathematical function that quantifies the performance of an ML model. Accuracy is a commonly used evaluation metric. |

| Loss | A *loss* function is a metric used to guide ML model training, by measuring the difference between predicted values and labels representing known (real) values. A frequently used loss function for regression is mean squared error; cross-entropy is a common one for classification. |
|---|---|
| *Data preparation* | *Data preparation* is the process of making raw data suitable and optimized for machine learning through the application of data transformation operations. |
| *Pipeline* | A *pipeline* is a sequence of steps applied within the machine learning workflow, which can include elements of data preparation, modeling, and evaluation. |

Table 5: Definitions of fundamental ML concepts relevant to data preparation

