# OpenReview forum: "Automated Data Preparation for Machine Learning: A Survey"
_DMLR — Accepted by DMLR_

### Review · Reviewer_UCK4 · 2025-09-22

**Recommendation:** 4
**Confidence:** 2

**Summary Of Contributions:**

A review paper - This one proposes taxonomy for automated data preparation, and maps existing AutoML systems and tools to it, rule-based vs search-based pipelines. It also highlights evaluation pitfalls and the absence of standard benchmarks, identifies data integration/cleaning and emerging LLM-assisted workflows.

**Strengths:**

Strengths:
Clear taxonomy
The discussion of model performance as a proxy for pipeline quality is concise and reasonable.
It differentiates static vs optimized data-prep in AutoML and adds semi-automated and fully automated
The gap between data-centric rhetoric and AutoML practice is well presented.

**Audience:**

Yes

**Broader Impact Concerns:**

No major concerns

**Claims And Evidence:**

Some evidence related to benchmarks is not clear. But if they address the weakness, it will be answered.

**Datasets And Benchmarks:**

There is not sufficient detail on collection/ how review was conducted/scoped. Weakness point 1.

**Extended Submissions:**

It meets the eligibility criteria.

**Limitations:**

Same as weakness

**Requested Changes:**

Please add Systematic Review Methods section.
You are describing categories but do not quantify prevalence trends
Maybe add a template for benchmarks?
Quantify risks and create a checklist accordingly so people can follow.

**Strengths And Weaknesses:**

Strengths:
Clear taxonomy
The discussion of model performance as a proxy for pipeline quality is concise and reasonable.
It differentiates static vs optimized data-prep in AutoML and adds semi-automated and fully automated
The gap between data-centric rhetoric and AutoML practice is well presented.

Weakness:
No Systematic Review Methods section.
You are describing categories but do not quantify prevalence trends
Maybe add a template for benchmarks?
Quantify risks and create a checklist accordingly so people can follow.

---

### Review · Reviewer_YQgE · 2025-09-23

**Recommendation:** 4
**Confidence:** 2

**Summary Of Contributions:**

Paper surveys and analyses current state of the data preparation for machine learning, especially concentrating on automating the data preparation process. The scope of the presentation is focused on the tabular data type of ML problems. The main contribution is the presentation of data preparation for ML as a whole, and surveying and synthesising, based on the literature, how to automated it. First, general data transformations, and pipelines of these transformation and hyper parameter optimisation, are examined from different perspectives. Second, based on former, different AutoML systems are presented and analysed in relation to different data preparation sub categories. Third, semi-automatic data preparation systems are surveyed and compared. Fourth, fully automated data preparation methodologies are introduced with comparative overview. General findings (along with combining the existing literature and systems) are that the most of the current autoML systems rely on ready-made input data and feature-level transformations, and only a few of the systems include pipeline optimisation capabilities whereas semi-automated approaches concentrate more on data manipulation and cleaning, overall. In the case of fully automated optimisation approaches, it is shown that there have been significant progress recently, but there are variety of different characteristics and satisfied requirements between systems.

**Strengths:**

The paper is generally well-written and organised, providing comprehensive survey and synthesis of current literature, techniques, and systems for automating data preparation. Main strength and contribution is to bring together, grouping, and unifying  different aspects of automated data preparation, missing from previous surveys. Rich set of literature and existing systems are analysed and compared, and general characteristics of different categories are discussed. Overall, a good paper that is relevant to broader research community.

**Audience:**

Yes

**Broader Impact Concerns:**

Broader Impact Statement is present in the paper. General concerns related to topic are discussed.

**Claims And Evidence:**

Most of the paper claims are supported. There are few issues (listed in the requested changes and limitations above) that could improve and clarify the evidence and better support the claims, if revised and adjusted.

**Datasets And Benchmarks:**

Submission does not include dataset or benchmark.

**Extended Submissions:**

To my knowledge, this is a new submission.

**Limitations:**

List of weaknesses and limitations
- Missing detailed alignment of the proposed work with previous surveys (comparison table could be added at the beginning)
- There is a bit "list" like writing in some of the sections (could be support with additional visualisation and graphs)
- Unclear description how different requirements are evaluated (e.g., in Table 3)
- Limited general discussion section of the findings at the end (including end-user perspectives)
- Research directions and challenges could be also discussed more detailed

**Requested Changes:**

There are few adjustments that could be consider to improve the work:
1. Table or some kind of more detailed comparison about the contributions in relation to other survey articles
2. There could be more detailed description of how system requirements are evaluated (i.e., which system/method satisfies the
requirement, when comparing fully automated approaches)
3. Discussion section considering summary of research opportunities and challenges
4. Based on the findings, end-user guidance of what solution to be selected for particular problems, could be added

**Strengths And Weaknesses:**

Strengths
- Generally well-written and organised manuscript
- Well-categorised methodological classes and the grouping of current techniques in the data preparation pipeline
- Unifying the data preparation for ML
- Rich set of literature surveyed and synthesised

Weaknesses
- Missing detailed and comprehensive discussion and comparison how of the proposed work with differs from previous surveys
- Focusing mostly on tabular input data types only
- Limited general discussions of the findings, supporting new research directions and challenges as well as guidance from end-user perspectives

---

### Review · Reviewer_JRT6 · 2025-09-24

**Recommendation:** 3
**Confidence:** 3

**Summary Of Contributions:**

The work presented here is a survey article of the state of play in automating the "data preparation pipeline" for machine learning applications, thereby extending "AutoML", i.e. automating the modelling side. The article seeks to define these terms from scratch and restricts attention to automated data preparation methods for tabular data. Its stated aim is to survey the existing body of knowledge, seeking to offer insights regarding scope, relevance, challenges and methodologies thus providing a platform for further exploration.

**Strengths:**

Please see the bullet point list above.
The paper's topic is relevant and would be of interest to the wider academic community. I find Figure 1 insightful.

**Audience:**

Yes

**Broader Impact Concerns:**

I have no broader impact concerns.

**Claims And Evidence:**

This is a survey article and most claims are well attributed. As highlighted in the weaknesses section, there are quite a few side remarks, even quotes which do not reflect the full context in which they were made, that carry strong opinions that might not be shared with all readers. This can be easily addressed though.

**Datasets And Benchmarks:**

not applicable here.

**Extended Submissions:**

NA

**Limitations:**

Please see above (section Strength and Weaknesses). As outlined above, I think major revision is needed. That said, the paper contains some interesting material (Table 1 for instance) that, imho, would help draw the attention of the academic community to this set of questions. In its current form, I do not think it can achieve this, but I would like to think that the authors can rescue their hard work and will be happy to revise my assessment if they do.

**Requested Changes:**

I think this paper needs some major revision to its field of application and its core review structuring methodology.
The paper's current concentration on tabular data is confusing, as this term is not properly defined and, in the way I read it, is then
overly restrictive to the point that it might even be not relevant to most deep learning related problems. Let me stress that there is nothing fundamentally wrong in not concentrating on deep learning method at all, but then most of the paper's content seems somewhat tied to it, as outlined in the Strengths and Weakness section above.
With regards to the underlying review methodology proposed to tackle this incredibly difficult field, I think this needs some further thinking.
When reading this review, I gained the impression, maybe wrongly, that the authors came at this from a real world systems design point of view. Here, data pipelines really are separate tools with separate software design principles underpinning them, that can be mapped straight into kubernetes deployment files or the equivalent of other orchestration tools. Maybe many of the paper's weaknesses I highlight can be addressed in using the "software 1.0" and "software 2.0" distinctions, i.e. between traditionally computer language defined algorithms and those trained through gradient descent. It seems to me the paper presupposes that the data preparation pipeline should live in the software 1.0 world and defining this explicitly might help its approach. But again, there might be other ways to fix this.

**Strengths And Weaknesses:**

**Strengths:***
- Interesting underlying question with potentially large impact.
- The article's prose is admirable rendering the article a good read.
- Table 1 is insightful.


**Weaknesses:**
- The survey's focus on tabular data makes the survey less relevant. Now, taken literally, nearly everything is tabular data.  Images are if you allow the columns to be successive pixel columns and the rows to be the corresponding vectors of columns of pixel values. RL trajectory rollouts in many cases are scalar tabular data as well. But the description and all considerations make it clear that the data treated here is likely meant to be spreadsheet collected clinical data, with scalar values and each column representing separate features of a different kind, although this is not well defined. Apart from a few towering examples, deep learning methods have so far failed to be really impactful in this area and random forest methods, e.g. implemented in Xboost, will be everyone's goto toolkit. The articles' focus would logically want to take data preparation methods for this class of algorithms into account, which differs from the work presented, as these methods are much more traditional requiring a lot less compute and are nearly a solved problem in terms of automation.
- I find the paper's well-intentioned attempt to properly define terms and processes overly restrictive, leading to it seemingly not applying to many interesting real world situations. Key examples are:
    1. Adopting the terms of Figure 1 in the paper (in quotation marks), most real world data collection is initiated by the desired "Real World insights" and prevailing solution approaches particularly in ML at the time of system design play a role in how the data collection design pans out. Example: Google Maps and other mapping companies started out as wanting a as real as possible digital representation of the world that can be searchable and useful (predict ETAs, say). At the time, modelling the world via satellite images, creating world wide road graphs, collecting probe data from cars to gauge the traffic condition on these road graphs determined the large scale data pipeline's design. Today, with deep learning methods on the rise, you can see the same phenomena in drug design, real time weather prediction, self-driving, etc. Start ups are being funded and, apart from on compute, a lot of the investment goes to collecting data in ways known to be the key ingredient in known novel algorithms and where it is clear that data currently does not exist. Prime examples here are RL algorithms (and roll-outs can be simple tabular data, too, in many real world examples) where some require "replay buffers" or example trajectories and extrapolate from them. Thus Figure 1 has (a not shown) problem which is really determined by the desired "Real World insights" and "data collection" and "problem modelling" are interdependent with the zeroth step in the paper's section 2.2 then being data collection design.
    2. If one takes the viewpoint of the last bullet point, this also sheds a new light on evaluating both the data quality and data preparation pipeline. The desired "Real World insights" and associated metrics are then the main driver which, as described, are model and solution dependent.
    3. To me, the intended definition of where a model starts and where data preparation transformations end is not well defined, leading to inconsistencies. Example: Suppose I have a trained GAN with a generator and discriminator. The generator could be seen as part of the data preparation pipeline, in fact it can be trained separately or in stages. Similarly with all neural networks where lower layers can be frozen and trained independently in stages and can be seen as part of the input data to the upper layers. In fact, cross-attention layers of a certain kind are known to be mathematically equivalent to essentially performing clustering, the being mentioned as a transformation method. The paper mentions tokenization, so BPE is considered as a data transformation, but does this then also now include learnt encodings and hence that part of a model if one wants to see the entire system as a large model? Similarly if methods like CLIP or other unsupervised methods are used to automatically align content, is that step part of the data preparation pipeline? The reason this matters for the paper is that its following sections on evaluation etc then become inconsistent.
    4.  The paper does not distinguish well between the loss functions used for training and the desired evaluation metrics to assess the overall outcome. The second sentence in 2.4 makes it clear when it says "[T]the main performance criterion is usually a chosen effective metric for the model (its loss function)". Many desired metrics are not differentiable or hard to apply in gradient descent training, hence these two concepts are usually different. For instance in many classification tasks cross-entropy is taken to be the loss function whereas the desired metric to judge the outcome is accuracy. This imprecision is also prevalent in the generalized formulation of the data pipeline optimization problem presented on page 14. The reason why this matters is that it is much simpler to formulate non-differentiable metrics for success than it is to design corresponding loss functions, hence a single "L" (if it is meant to be differentiable and trainable) would be hard to find. Maybe this L should be replaced by one or several non-differentiable quality metrics, some of which might also be data dependent. This is not just a remark to make a theoretical section more tight, but has real world implications even to the structure of this review. Hunting for good loss functions that help a metric and allow for simpler models to be used (one can argue that RL is necessary only because cumulative rewards are not a differentiable signal for direct gradient descent) is a key step both AutoML and automatic data pipelining will need to address.
    5. Sticking with the generalized formulation on page 14, should there not be a real world utility component (such as wall time, compute limit and other resource constraints) associated with this formulation? In drug design, say, this formulation could then just be "great, let's just list those 10^60 possible small molecules and process them with a few As ..." which is infeasible.
- There are quite a few rather strong opinions buried in parts of the text. The paper states in general that "data quality sets an upper bound on model performance with regards to reality" attributing this to a given quote without qualifying the latter's constraint to that statement.
For instance, work on the decision transformer have shown that in RL optimal trajectories can be gleaned from a large collection of suboptimal ones. In the introduction, the paper states that "[M]machine learning heavily relies on human expertise, ..." without qualifying this statement, whereas the article's own work shows that, in few settings, this not to be the case.
- I appreciated the article's stated attempt to combine notions and terms in the database field with those in ML. Yet, even after attentive reading, all I can find is the use of the acronym ETL and other commonly used terms without clarifying what the different connotations are in these respective fields. To my knowledge, in DB circles, data accuracy and quality are defined more in terms of type setting and data formats than maybe in terms of data science terms.